# Spectral Guidance for Flexible and Efficient Control of Diffusion Models

**Gabriel Moreira** [1 2]   **Manuel Marques** [1]   **João Paulo Costeira** [1]   **Chenyan Xiong** [2]

## Abstract

We introduce Spectral Guidance, a framework for controlling diffusion models by leveraging the intrinsic geometry of the generative process. As data is progressively corrupted by noise, only a small number of features remain informative for control. We characterize them as the singular functions of a conditional expectation operator and show that they can be learned via a self-supervised objective. Once recovered, this basis enables the projection of arbitrary guidance signals, such as labels, CLIP embeddings, or masks, directly onto the sampling trajectory. This approach allows for stable, high-fidelity control without retraining or denoiser backpropagation during sampling. Empirically, we improve conditional accuracy on CIFAR-10 by 37 percentage points over the strongest training-free baseline while offering $4\times$ faster sampling. Moreover, the same representations that support label and CLIP guidance also enable spatial control, such as mask-based guidance, without auxiliary models. Finally, our framework reveals a phase transition in the generative process, pinpointing the optimal time window for effective guidance.

## 1. Introduction

Generative modeling has seen tremendous progress with diffusion-based approaches (Sohl-Dickstein et al., 2015), which now produce high-fidelity samples across images, audio, and other modalities (Ho et al., 2020; Song et al., 2020b; Rombach et al., 2022). These models operate by reversing a progressive corruption process, gradually transforming structured data into noise. While the forward and reverse dynamics of generation are well-understood, the utility of these models hinges on *guidance*: sampling ac-

cording to user specifications, which can be in the form of labels (Dhariwal & Nichol, 2021), text prompts (Saharia et al., 2022), or custom objectives (Zhang et al., 2023).

The key challenge is how to impose such guidance in practice. Successful mainstream approaches rely on training models to be conditional from the outset (Dhariwal & Nichol, 2021; Ho & Salimans, 2022), so that desired guidance can be injected directly during sampling. This strategy yields strong and stable control, but tightly couples the model to a fixed set of conditions and requires retraining or additional models whenever the specification changes. An alternative is to start from an unconditional model and enforce the desired behavior only at sampling time by optimizing a user-defined objective (Chung et al., 2022; Ye et al., 2024). While this offers greater flexibility, it typically requires differentiating through the denoiser during sampling and approximating an intractable posterior distribution. In practice, this leads to higher computational cost and often unstable control, especially for complex objectives.

In this work, we propose Spectral Guidance, a framework that enables flexible control by leveraging the intrinsic structure of the diffusion process. As data is gradually corrupted by noise, fine-grained details are lost while coarse semantic features persist. We show that these features form a natural, low-dimensional coordinate system that tracks how information propagates through the diffusion. By learning this representation, we can project arbitrary guidance objectives, from simple labels to masks, directly into the generative trajectory. This allows for stable control without the need for task-specific retraining or denoiser gradients.

Our technical approach is based on a low-rank approximation of the conditional expectation operator across diffusion timesteps. Because high-frequency details are progressively destroyed, the leading singular functions of this operator form a time-indexed low-dimensional basis that captures persistent axes of variation over diffusion time. We show that a self-supervised learning (SSL) objective with orthogonality constraints (Bardes et al., 2021) is a variational estimator for the operator's leading singular functions, with independently diffused views of the same sample acting as augmentations. Once these representations are learned, guidance reduces to a simple linear projection onto this basis.

This formulation provides an efficient and flexible guid-

[1]Institute for Systems and Robotics, Instituto Superior Técnico, Lisbon, Portugal [2]Carnegie Mellon University, Pittsburgh, PA, USA. Correspondence to: Gabriel Moreira <gmoreira@cs.cmu.edu>.

*Proceedings of the 43rd International Conference on Machine Learning*, Seoul, South Korea. PMLR 306, 2026. Copyright 2026 by the author(s).

ance mechanism. It removes the need for backpropagation through the denoiser during sampling, requiring only a one-time training to learn the intrinsic coordinates of the diffusion process. These representations reveal which features are recoverable and when guidance is effective. Further, being task-agnostic, they support arbitrary downstream control objectives without retraining.

Empirically, we achieve consistent gains across label, attribute, and CLIP guidance; notably, on CIFAR-10, we surpass the strongest training-free baseline by 37 percentage points in accuracy while improving FID and sampling $4\times$ faster. In addition, our representations generalize to spatial control like mask-guided generation without auxiliary models. Finally, they reveal a spectral phase transition that pinpoints the optimal time window for effective guidance.

Our contributions are summarized as follows:

- We propose Spectral Guidance, which reframes diffusion guidance as a projection onto a coordinate system aligned with the generative dynamics, enabling flexible control without retraining the diffusion model.

- We introduce an SSL objective that estimates the spectral decomposition of the diffusion operator. This representation yields a lightweight guidance algorithm, decoupled from the gradients of the denoiser.

- We outperform training-free baselines by over 37 percentage points in accuracy and $4\times$ in speed, while enabling complex controls like mask guidance without auxiliary models. Further, our representations reveal a phase transition during the reverse process that aligns with the optimal time window for effective guidance.

Code available at https://github.com/gabmoreira/spectralguidance

## 2. Related Work

We review key advances in conditional generation below.

**Diffusion guidance.** To introduce conditioning or amplify specific signals during sampling, Classifier Guidance (CG) (Dhariwal & Nichol, 2021) employs an external classifier trained on noisy data, using its gradients to steer the generative trajectory. Classifier-Free Guidance (CFG) (Ho & Salimans, 2022) alleviates the need for external classifiers by jointly training a conditional and unconditional model, subsequently interpolating their score estimates during sampling. While CFG has become the *de facto* standard for modern architectures, ranging from Stable Diffusion (Rombach et al., 2022) to Flow Matching models (Lipman et al., 2022), it typically applies a constant guidance scale, remaining agnostic to the dynamics of the diffusion process.

For a comprehensive survey, we refer to Zhan et al. (2024). To address the limitations of static guidance, recent works (Koulischer et al., 2025; Kynkäänniemi et al., 2024) leverage insights into diffusion dynamics and phase transitions (Handke et al., 2025; Raya & Ambrogioni, 2023) to target time windows where features are most controllable, thereby optimizing generation quality.

**Training-free guidance.** A distinct line of research focuses on controlling pre-trained diffusion models via external loss functions, in a plug-and-play fashion, eliminating the need for retraining. Initially developed for inverse problems (Kawar et al., 2021; Choi et al., 2021; Kawar et al., 2022), this paradigm includes Diffusion Posterior Sampling (DPS) (Chung et al., 2022), which guides sampling with the gradients of a loss function evaluated on a point-estimate of the posterior distribution. For general control, Loss-Guided Diffusion (LGD) (Song et al., 2023) refines DPS by estimating the conditional expectation via Monte Carlo sampling, while MPGD (He et al., 2024) leverages the manifold hypothesis to constrain guidance to low-dimensional data manifolds. Universal Guidance for Diffusion (UGD) (Bansal et al., 2023) and FreeDoM (Yu et al., 2023) further strengthen guidance through iterative "time-travel" strategies and adaptive schedules across diffusion timesteps. More recently, Training-Free Guidance (TFG) (Ye et al., 2024) unifies many of these approaches under a common guidance algorithm.

**Editing directions in diffusion models.** NoiseCLR (Dalva & Yanardag, 2024) discovers interpretable directions in the noise space of pre-trained diffusion models via a contrastive self-supervised objective; while it shares the use of SSL-style training, it targets latent-space *editing* rather than information-preserving structure. More closely related, Park et al. (2023) and Chen et al. (2024) use the spectral decomposition of the denoiser's Jacobian as a post hoc tool for identifying semantic editing directions.

In contrast, we avoid training task-dependent conditional denoisers and relying on point estimates. Instead, we propose to guide unconditional models by mapping guidance signals onto the diffusion model's spectral coordinates.

## 3. Preliminaries

**Diffusion models.** Let $X_0 \sim p_0$ be a data random variable with support $\mathcal{X}$. Denoising Diffusion Probabilistic Models (DDPMs) (Ho et al., 2020) define a forward process that gradually perturbs $X_0$ into Gaussian noise using a variance schedule $\{\alpha_t\}_{t=1}^T$, where $\alpha_t \in (0,1)$ and $\bar{\alpha}_t := \prod_{s=1}^t \alpha_s$. This process allows for direct sampling of any noisy latent $x_t$ at timestep $t$ via

$$p_t(x_t \mid x_0) = \mathcal{N}\left(x_t; \sqrt{\bar{\alpha}_t}x_0, (1-\bar{\alpha}_t)I\right). \quad (1)$$

To reverse this process, a neural network $\epsilon_\theta(x_t, t)$ is trained to predict the noise $\epsilon$ added to $x_0$. This training objective is equivalent to denoising score matching (Song & Ermon, 2019), as the optimal denoiser is related to the score of the marginal distribution by

$$\nabla_{x_t} \log p_t(x_t) = -\frac{\epsilon_\theta(x_t, t)}{\sqrt{1 - \bar{\alpha}_t}}. \tag{2}$$

While the original DDPM formulation requires a stochastic Markov chain, Denoising Diffusion Implicit Models (DDIMs) (Song et al., 2020a) provide a faster, non-Markovian alternative, via the update rule

$$x_{t-1} = \sqrt{\bar{\alpha}_{t-1}} \left( \frac{x_t - \sqrt{1 - \bar{\alpha}_t}\epsilon_\theta(x_t, t)}{\sqrt{\bar{\alpha}_t}} \right)$$
$$+ \sqrt{1 - \bar{\alpha}_{t-1} - \sigma_t^2}\epsilon_\theta(x_t, t) + \sigma_t\varepsilon, \tag{3}$$

where $\varepsilon \sim \mathcal{N}(0, I)$.

**Diffusion guidance.** The practical utility of the generative reverse-time process depends on the ability to guide the unconditional trajectory in Eq. (3), given a conditioning signal $y$ (such as a class label, text prompt, or task objective). This corresponds to sampling from the conditional distribution $p(x_0 \mid y)$. By Bayes' rule we can decompose the conditional score as

$$\nabla_{x_t} \log p_t(x_t \mid y) = \nabla_{x_t} \log p_t(x_t) + \nabla_{x_t} \log p_t(y \mid x_t).$$

While $\nabla_{x_t} \log p_t(x_t)$ is provided by the unconditional model, the guidance term $\nabla_{x_t} \log p_t(y \mid x_t)$ requires a time-dependent predictor (Dhariwal & Nichol, 2021),

$$p_t(y \mid x_t) = \mathbb{E}_{X_0 \sim p_t(\cdot|x_t)}[p(y \mid X_0)]. \tag{4}$$

Hence, this approach restricts control of the generation to the conditioning signal $y$ used for training $p_t(y \mid x_t)$.

**Training-free guidance.** In contrast to training-based approaches, training-free guidance aims at steering the unconditional trajectory in Eq. (3) using any clean data signal $p(y \mid x_0)$ by estimating $p_t(y \mid x_t)$ during sampling. As Eq. (4) is generally intractable, methods such as DPS (Chung et al., 2022) rely on the denoiser's point estimate of the posterior mean $\hat{x}_0(x_t) \approx \mathbb{E}[X_0 \mid X_t = x_t]$,

$$\mathbb{E}_{X_0 \sim p_t(\cdot|x_t)}[p(y \mid X_0)] \approx p(y \mid \hat{x}_0(x_t)). \tag{5}$$

This substitution is exact only when $p(y \mid x_0)$ is affine in $x_0$, which rarely holds for semantic tasks. At large noise levels the posterior mean may even lie outside the data manifold (He et al., 2024), leading to misaligned gradients. Further, differentiating $\hat{x}_0(x_t)$ requires backpropagating through the denoiser at every step, which is computationally expensive and prone to vanishing gradients.

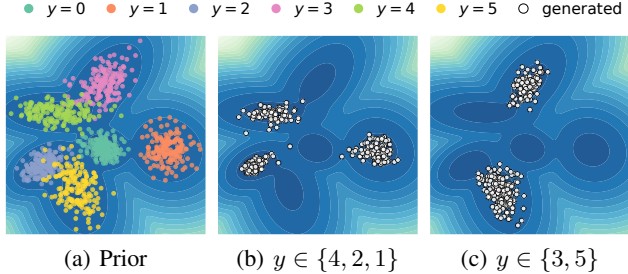

(a) Prior      (b) $y \in \{4, 2, 1\}$      (c) $y \in \{3, 5\}$

*Figure 1.* **Spectral Guidance on a Gaussian mixture prior.** Contours depict $\log p_0(x_0)$. **(a)** Prior samples colored by component. **(b, c)** Samples generated by spectral guidance (white) conditioned on different label subsets, using a fixed set of $K = 30$ spectral modes. The same features enable sampling from $p(x_0 \mid y \in \mathcal{Y})$ for arbitrary conditioning sets $\mathcal{Y}$.

## 4. Spectral Representation for Guidance

A key observation motivates our approach: as diffusion noise increases, information about the data is progressively destroyed, and only a small number of features remain recoverable. Consequently, at each diffusion timestep, there exists a low-dimensional set of intrinsic directions along which guidance can effectively act. We propose to guide samples along these diffusion-stable directions (Fig. 1).

We can view the expectation in Eq. (4) as the action of an operator from the clean data space $\mathcal{H}_0 = L^2(p_0)$ to the noisy data space $\mathcal{H}_t = L^2(p_t)$. This yields $p_t(y \mid x_t) = (T_t p(y \mid \cdot))(x_t)$, where $T_t : \mathcal{H}_0 \to \mathcal{H}_t$ is the conditional expectation of a function of the clean data $f(x_0)$ given $x_t$,

$$(T_t f)(x_t) := \mathbb{E}_{X_0 \sim p_t(\cdot|x_t)}[f(X_0)]. \tag{6}$$

This operator retains components of $f$ that are recoverable from the noisy observation $x_t$. To identify them, we consider the covariance operator $T_t T_t^*$, where the adjoint operator $T_t^* : \mathcal{H}_t \to \mathcal{H}_0$ corresponds to the forward process

$$(T_t^* g)(x_0) := \mathbb{E}_{X_t \sim p_t(\cdot|x_0)}[g(X_t)]. \tag{7}$$

The covariance operator $T_t T_t^*$ represents a round-trip information bottleneck, smoothing features of noisy data via the adjoint and then reconstructing them,

$$(T_t T_t^* f)(\tilde{x}_t) = \mathbb{E}_{X_0 \sim p_t(\cdot|\tilde{x}_t)} \left[ \mathbb{E}_{X_t \sim p_t(\cdot|X_0)}[f(X_t)] \right]. \tag{8}$$

The principal modes of $T_t T_t^*$ correspond to the directions in the noisy data that survive the round-trip. These are the intrinsic coordinates along which guidance can act. At high noise levels, only a few modes remain significant, allowing $T_t$ to be approximated by a low-rank decomposition.

### 4.1. Spectral Decomposition

If $p_0$ has compact support, $T_t T_t^*$ is compact (App. A.2, Theorem A.4) and self-adjoint. Thus, it admits an eigenvalue decomposition and $T_t$ a singular value decomposition

*i.e.*, there exist singular values $\{\sigma_{t,k}\}_{k=1}^\infty$, and orthonormal bases $\{\psi_{t,k}\}_{k=1}^\infty \subset \mathcal{H}_0$ and $\{\phi_{t,k}\}_{k=1}^\infty \subset \mathcal{H}_t$ such that

$$(T_t f)(x_t) = \sum_{k=1}^\infty \sigma_{t,k} \phi_{t,k}(x_t) \mathbb{E}_{p_0}[f(X_0)\,\psi_{t,k}(X_0)], \quad (9)$$

with $\sigma_{t,1} = 1$ and $\phi_{t,1} \equiv \psi_{t,1} \equiv 1$ corresponding to the constant mode (App. A.3, Proposition A.5). The right singular functions $\psi_{t,k} \in \mathcal{H}_0$ capture clean-space features preserved by diffusion. The left singular functions $\phi_{t,k} \in \mathcal{H}_t$ represent their optimal reconstructions from noisy data, corresponding to eigenfunctions of the covariance operator $T_t T_t^*$. The singular values $\sigma_{t,k}$ quantify robustness: large $\sigma_{t,k}$ indicate modes that are recoverable from noise while smaller $\sigma_{t,k}$ correspond to modes lost during diffusion.

## 4.2. Spectral Guidance

The SVD in Eq. (9) gives a natural decomposition of the posterior expectation of any guidance signal into basis functions $\{\phi_{t,k}\}_{k \geq 1}$, determined by the unconditional diffusion process, weighted by guidance-dependent coefficients $c_{t,k}$.

> **Proposition 4.1.** *For any $h \in \mathcal{H}_0$, its conditional expectation at time $t$ admits the expansion*
>
> $$\mathbb{E}_{X_0 \sim p_t(\cdot | x_t)}[h(X_0)] = \sum_{k=1}^\infty c_{t,k} \phi_{t,k}(x_t), \quad (10)$$
>
> *where $c_{t,k} := \mathbb{E}_{X_0, X_t}[h(X_0)\,\phi_{t,k}(X_t)].$*

We approximate the series in Eq. (10) by truncating it to its leading $K$ terms. This low-rank approximation incurs an $L^2(p_t)$ error bounded by $\sigma_{t,K+1}^2 \|h\|_{p_0}^2$ (App. A.4, Proposition A.11). However, the diffusion process ensures that

$$\sigma_{t,k}^2 \leq \mathbb{E}_{p_0}\left[\chi^2(p_t(\cdot \mid X_0) \,\|\, p_t)\right], \quad k \geq 2 \quad (11)$$

where $\chi^2$ denotes the chi-squared divergence. Because both $p_t(x_t \mid x_0)$ and $p_t(x_t)$ converge to the standard normal as $\bar{\alpha}_t \to 0$, the right-hand side vanishes, forcing every singular value beyond the first to zero (Proposition A.7). Thus, at high noise levels, the spectrum concentrates on a few leading modes and the truncated expansion closely approximates the true posterior expectation of the guidance signal.

## 4.3. Learning the Diffusion Spectrum

As $\phi_{t,k} \in \mathcal{H}_t$ are eigenfunctions of the covariance operator $T_t T_t^*$, they maximize the correlation between diffused views $(x_t, \tilde{x}_t) \sim \zeta$ of the same clean sample $x_0$, with

$$\zeta(x_t, \tilde{x}_t) := \int_\mathcal{X} p_t(x_t \mid x_0) p_t(\tilde{x}_t \mid x_0) p_0(x_0)\,dx_0. \quad (12)$$

This allows us to learn the leading left singular subspace $\mathrm{span}\{\phi_{t,k}\}_{k=2}^{K+1}$ of $T_t$ using an SSL objective.

> **Theorem 4.2** (Variational characterization). *Let $f = (f_1, \ldots, f_K)^\top$ with $\mathbb{E}_{p_t}[f] = 0$. Define the covariance*
>
> $$\boldsymbol{\Sigma}_t(f) := \mathbb{E}_{p_t}\left[f(X_t)\,f(X_t)^\top\right] \succ 0 \quad (13)$$
>
> *and the cross-covariance*
>
> $$\mathbf{C}_t(f) := \mathbb{E}_\zeta\left[f(X_t)\,f(\tilde{X}_t)^\top\right]. \quad (14)$$
>
> *Then,*
>
> $$\max_f\ \mathrm{Tr}\!\left(\mathbf{C}_t(f)\,\boldsymbol{\Sigma}_t(f)^{-1}\right) = \sum_{k=2}^{K+1} \sigma_{t,k}^2 \quad (15)$$
>
> *and any maximizer $f^\star$ satisfies $\mathrm{span}\{f_k^\star\}_{k=1}^K = \mathrm{span}\{\phi_{t,k}\}_{k=2}^{K+1}$.*

**Connection to SSL objectives.** SSL methods such as VI-CReg (Bardes et al., 2021) and Barlow Twins (Zbontar et al., 2021) rely on hand-crafted augmentations (cropping, color jittering) to define invariance. We replace these heuristics with the diffusion process itself. The optimization problem in Eq. (15) is the Rayleigh-Ritz characterization of the top-$K$ eigenspace, equivalent to a Kernel PCA with the kernel defined by the joint distribution $\zeta$. Maximizing the diagonal of $\mathbf{C}_t(f)$ enforces that the representation is invariant under the noise process, while the whitening transformation $\boldsymbol{\Sigma}_t(f)^{-1}$ prevents collapse into a constant solution.

## 5. Learning to Guide

The complete Spectral Guidance algorithm consists of an offline stage and a sampling stage; the latter augments an unconditional diffusion sampler with a guidance signal applied at a designated set of diffusion timesteps $\mathcal{T}$. We first describe how we learn the leading left singular functions of the conditional expectation operator. We then show how they are used for guidance.

### 5.1. Learning Singular Functions

As the constant mode is known *a priori* ($\phi_{t,1} \equiv 1$), we parameterize the next $K$ left singular functions of $T_t$ by a ResNet (He et al., 2016) with time-modulation using FiLM layers (Perez et al., 2018). The network $f_\phi : \mathcal{X} \times \mathbb{R}_{>0} \to \mathbb{R}^K$ receives a noisy sample and its timestep as input. Given a mini-batch $\{x_0^{(i)}\}_{i=1}^B \sim p_0$ and a timestep $t \in \mathcal{T}$, we draw coupled noisy samples $(x_t, \tilde{x}_t)$ according to

$$x_t^{(i)},\ \tilde{x}_t^{(i)} \stackrel{\text{i.i.d.}}{\sim} p_t\!\left(\cdot \mid x_0^{(i)}\right), \quad (16)$$

where $p_t(x_t \mid x_0)$ is the forward diffusion process. We evaluate $f_\phi$ on these two views, yielding

$$\mathbf{Z} = f_\phi(x_t, t), \quad \tilde{\mathbf{Z}} = f_\phi(\tilde{x}_t, t) \in \mathbb{R}^{B \times K}. \quad (17)$$

**Algorithm 1** Training loss for $f_\phi$

---

**Input:** Prior distribution $p_0$, diffusion schedule $\{\bar{\alpha}_t\}_{t=1}^T$
Mini-batch $\{x_0^{(i)}\}_{i=1}^B \sim p_0$
Sample timestep $t \sim \text{Uniform}(\mathcal{T})$
Sample noise $\{\epsilon^{(i)}\}_{i=1}^B, \{\tilde{\epsilon}^{(i)}\}_{i=1}^B \sim \mathcal{N}(\mathbf{0}, \mathbf{I})$
$x_t^{(i)} \leftarrow \sqrt{\bar{\alpha}_t} x_0^{(i)} + \sqrt{1 - \bar{\alpha}_t} \epsilon^{(i)}$
$\tilde{x}_t^{(i)} \leftarrow \sqrt{\bar{\alpha}_t} x_0^{(i)} + \sqrt{1 - \bar{\alpha}_t} \tilde{\epsilon}^{(i)}$
$\mathbf{Z} \leftarrow \text{sg}(f_\phi(x_t, t)), \tilde{\mathbf{Z}} \leftarrow f_\phi(\tilde{x}_t, t)$
$\boldsymbol{\mu} \leftarrow \text{col-mean}(\mathbf{Z})$
$\hat{\boldsymbol{\Sigma}} \leftarrow (\mathbf{Z} - \boldsymbol{\mu})^\top (\mathbf{Z} - \boldsymbol{\mu})/(B - 1)$
$\mathbf{V}, \boldsymbol{\Lambda} \leftarrow \text{eigh}(\hat{\boldsymbol{\Sigma}})$
$\mathbf{W} \leftarrow \mathbf{V}(\boldsymbol{\Lambda} + \xi\mathbf{I})^{-1/2}$
$\mathbf{Z}^w \leftarrow (\mathbf{Z} - \boldsymbol{\mu})\mathbf{W}, \tilde{\mathbf{Z}}^w \leftarrow (\tilde{\mathbf{Z}} - \boldsymbol{\mu})\mathbf{W}$
**Return:** $L = -\text{Tr}\big((\mathbf{Z}^w)^\top \tilde{\mathbf{Z}}^w\big)/(K(B - 1))$

---

**Whitening.** We implement the whitening transformation $\boldsymbol{\Sigma}_t(f)^{-1}$ in Theorem 4.2 via the eigendecomposition of the empirical covariance. Let $\boldsymbol{\mu} \in \mathbb{R}^K$ denote the column-mean of $\mathbf{Z}$ and write

$$\hat{\boldsymbol{\Sigma}} := \frac{1}{B - 1}(\mathbf{Z} - \boldsymbol{\mu})^\top (\mathbf{Z} - \boldsymbol{\mu}) = \mathbf{V}\boldsymbol{\Lambda}\mathbf{V}^\top. \qquad (18)$$

We define the whitening matrix as

$$\mathbf{W} := \mathbf{V}(\boldsymbol{\Lambda} + \xi\mathbf{I})^{-1/2}, \qquad (19)$$

where $\xi > 0$ is a small ridge hyperparameter ensuring $\mathbf{W}$ is well defined when $\hat{\boldsymbol{\Sigma}}$ is rank-deficient.

**Loss.** Let $\mathbf{Z}^w := (\mathbf{Z} - \boldsymbol{\mu})\mathbf{W}$ and $\tilde{\mathbf{Z}}^w := (\tilde{\mathbf{Z}} - \boldsymbol{\mu})\mathbf{W}$ denote the whitened views. A Monte Carlo approximation of the objective in Eq. (15) yields the loss

$$L = -\frac{1}{K(B - 1)} \text{Tr}\left((\mathbf{Z}^w)^\top \tilde{\mathbf{Z}}^w\right), \qquad (20)$$

which is equivalent to the objective in Theorem 4.2, up to the ridge regularization. Following common SSL practice, we apply a stop-gradient (sg) to $\mathbf{Z}$ (which also detaches the whitening statistics $\boldsymbol{\mu}$ and $\mathbf{W}$), which empirically stabilizes training. The full procedure, which is independent of the U-Net denoiser, is given in Algorithm 1.

**Reference statistics.** The training loss whitens the output of $f_\phi$ with batch statistics recomputed at every step. To obtain a deployable estimator that can be evaluated on a single noisy sample and produce population-whitened features, we compute, post-training, a whitening transformation per timestep on a reference set $\mathcal{D}_{\text{ref}} = \{x_0^{(i)}\}_{i=1}^M$. For each $t \in \mathcal{T}$, we draw noisy versions $x_t^{(i)} \sim p_t(\cdot \mid x_0^{(i)})$ and encode them with $f_\phi$ to obtain a feature matrix $\mathbf{Z}_t \in \mathbb{R}^{M \times K}$. We set $\boldsymbol{\mu}_t$ to its column-mean and compute $\mathbf{W}_t$ from

Eq. (19) on the centered features. The whitened network, with the constant mode appended, is then

$$f_\phi^w(x_t, t) := \begin{bmatrix} 1 & \left(f_\phi(x_t, t) - \boldsymbol{\mu}_t\right)\mathbf{W}_t \end{bmatrix}, \qquad (21)$$

and we cache its evaluation on the reference set,

$$\boldsymbol{\Phi}_t := \begin{bmatrix} 1 & (\mathbf{Z}_t - \boldsymbol{\mu}_t)\mathbf{W}_t \end{bmatrix} \in \mathbb{R}^{M \times (K+1)}, \qquad (22)$$

for use in guidance. By construction, $\boldsymbol{\Phi}_t$ has orthogonal columns; on a fresh sample $x_t$, the outputs of $f_\phi^w$ are approximately zero-mean with identity covariance under $p_t$.

### 5.2. Guidance

We consider clean-data guidance signals $h(x_0) \in \mathbb{R}^{D_h}$. Examples include the probability $p(y \mid x_0)$ of a target class $y$ ($D_h = 1$), a CLIP image embedding ($D_h = D_{\text{CLIP}}$) or a flattened binary segmentation mask over the $W \times H$ pixel grid ($D_h = WH$).

**Guidance coefficients.** To guide generation, we first estimate the spectral coefficients $c_{t,k} \in \mathbb{R}^{D_h}$ from Proposition 4.1. Evaluating $h$ on the clean samples of $\mathcal{D}_{\text{ref}}$, we construct $\mathbf{H} \in \mathbb{R}^{M \times D_h}$ and combine it with the cached reference matrix $\boldsymbol{\Phi}_t$ from Eq. (22) to obtain the Monte Carlo estimate

$$\hat{\mathbf{c}}_t = \frac{1}{M}\boldsymbol{\Phi}_t^\top \mathbf{H} \in \mathbb{R}^{(K+1) \times D_h}, \qquad (23)$$

whose $k$-th row $\hat{c}_{t,k} \in \mathbb{R}^{D_h}$ estimates the coefficient $c_{t,k}$.

**Sampling.** Algorithm 2 summarizes the sampling stage, which consists of a standard DDIM step followed by computing and applying a guidance vector $g$ with strength $\kappa$. For a noisy sample $x_t$, truncating the spectral expansion of Proposition 4.1 to the leading $K + 1$ terms yields the estimate $\hat{\mathbf{c}}_t^\top f_\phi^w(x_t, t) \approx \mathbb{E}_{X_0 \sim p_t(\cdot \mid x_t)}[h(X_0)]$. We define the guidance vector $g(x_t)$ as

$$g(x_t) = \nabla_{x_t}\mathcal{L}\left(\hat{\mathbf{c}}_t^\top f_\phi^w(x_t, t)\right), \qquad (24)$$

for a guidance-dependent loss function $\mathcal{L}$. If $h(x_0)$ is a class probability, $\hat{\mathbf{c}}_t^\top f_\phi^w(x_t, t) \approx p_t(y \mid x_t)$ and we use the log-likelihood. Since the truncated series approximation of the density may locally violate positivity, we set $\nabla_{x_t}\mathcal{L}(z) = (\nabla_{x_t}z)/z$. For CLIP guidance, Eq. (23) yields the expected CLIP embedding of the clean image. We employ the cosine similarity with the normalized text embedding $\mathbf{e}_{\text{text}}$ as $\mathcal{L}(\mathbf{z}) = \mathbf{z}^\top \mathbf{e}_{\text{text}}/\|\mathbf{z}\|$. For mask guidance, Eq. (23) yields the expected clean mask and we set $\mathcal{L}(\mathbf{z}) = -\|\mathbf{z} - \mathbf{z}_{\text{target}}\|^2$, for a target mask $\mathbf{z}_{\text{target}}$.

**Complexity comparison.** Table 1 provides a complexity comparison between Spectral Guidance, CG and TFG.

---

**Algorithm 2** Spectral Guidance

---

**Input:** Timesteps $\mathcal{T}$; Strength $\kappa$; Denoiser $\epsilon_\theta$; DDIM scheduler $\mathcal{S}$; Coefficients $\{\hat{\mathbf{c}}_t\}_{t\in\mathcal{T}}$; Pre-trained $f_\phi$
**For** $t$ **in** reverse($\mathcal{T}$):
    Predict noise $\epsilon \leftarrow \epsilon_\theta(x,t)$
    Denoising step $x \leftarrow \mathcal{S}(\epsilon, x, t)$
    Guidance $g \leftarrow \nabla_x \mathcal{L}\left(\hat{\mathbf{c}}_t^\top f_\phi^w(x,t)\right)$
    Update latent $x \leftarrow x + \kappa\sqrt{1 - \bar{\alpha}_t}\, g$
**Output:** Final sample $x$

---

Spectral Guidance amortizes computational cost by shifting the heavy optimization to a one-time offline phase, avoiding the $\mathcal{O}(N_{\text{rec}} \cdot (\nabla_x \epsilon_\theta + \nabla_x f_{\text{cls}}))$ bottleneck of training-free methods that require backpropagating through the denoiser and classifier. By using a lightweight network $f_\phi$ (16M parameters vs. denoiser's 114M on CelebA-HQ), the per-step inference overhead is reduced to a shallow gradient $\nabla_x f_\phi$, with per-step latency comparable to classifier guidance.

## 6. Experiments

We evaluate Spectral Guidance along four axes: (i) its effectiveness and flexibility relative to training-free guidance baselines across categorical, text, and spatial conditioning tasks (§6.2); (ii) the sensitivity of guidance to the spectral rank $K$ and guidance strength $\kappa$ (§6.3); (iii) the connection between the spectrum of $T_t T_t^*$ and the temporal window in which guidance is most effective (§6.4); and (iv) sampling efficiency (§6.5).

### 6.1. Experimental Setup

**Models and data.** We evaluate conditional image generation on CIFAR-10 (Krizhevsky et al., 2009), CelebA-HQ (Karras et al., 2017), and ImageNet (Deng et al., 2009), using unconditional DDPM U-Nets with 36M (CIFAR-10), 114M (CelebA-HQ), and 121M (ImageNet) parameters. All experiments use a DDIM sampler. The spectral networks $f_\phi$ are time-conditioned ResNets (He et al., 2016) with FiLM modulation (Perez et al., 2018). The output dimensions are set to $K = 512$ on CIFAR-10 and CelebA-HQ, and $K = 2000$ on ImageNet. The $f_\phi$ networks are substantially lighter than the corresponding generative U-Nets: 9M parameters on CIFAR-10, 16M on CelebA-HQ, and 91M on ImageNet. Full details are provided in Appendix C.1.

**Baselines.** We compare our approach against state-of-the-art training-free guidance methods: DPS (Chung et al., 2022), LGD (Song et al., 2023), FreeDoM (Yu et al., 2023), UGD (Bansal et al., 2023), MPGD (He et al., 2024), and TFG (Ye et al., 2024), using the implementation and hyperparameters provided in the TFG framework.

**Tasks.** We evaluate three conditioning modalities.

- **Categorical.** We guide toward the 10 classes of CIFAR-10, combinations of attributes on CelebA-HQ (Gender + Age and Gender + Hair color), and 4 classes of ImageNet. The clean-data signal $h(x_0)$ is the one-hot label available with each dataset; baselines backpropagate through external classifiers.

- **Text (CLIP).** On CelebA-HQ, we evaluate generation conditioned on 15 text prompts *e.g.*, *"a young woman wearing sunglasses"*. In our method, $h(x_0)$ is the CLIP embedding of the clean image; baselines backpropagate cosine similarity through CLIP's image encoder.

- **Mask.** We evaluate mask guidance on CelebA-HQ by setting the clean-data signal $h(x_0)$ of our framework to a hair-segmentation mask. We guide all models using the gradients of the MSE between the target mask and the clean mask estimate; baselines backpropagate through an external face-parsing model.

Crucially, our approach relies on the same spectral representations $\{\Phi_t\}_{t\in\mathcal{T}}$ from Eq. (22) to support all three tasks. Only the guidance signal $h$ is swapped at sampling time.

**Metrics.** We report fidelity and guidance validity. Fidelity is measured via intra-class FID (Heusel et al., 2017) against target-class training images on CIFAR-10 and ImageNet, and via the log Kernel Inception Distance (log KID) on CelebA-HQ. For categorical guidance, validity is the top-1 accuracy of a pre-trained classifier on generated samples. For text guidance, we measure image–prompt alignment with VQAScore (Lin et al., 2024) (range $[0, 1]$, higher is better), using `llava-v1.5-7b`. For mask guidance, we report the Intersection-over-Union (IoU) between the target hair mask and the mask predicted by an independent segmentation model.

### 6.2. Overall Results

Table 2 reports validity and fidelity across all guidance tasks. Spectral Guidance attains the highest validity in every setting, with competitive fidelity on most tasks.

On CIFAR-10, Spectral Guidance reaches 89.4% accuracy, 37 percentage points above the strongest baseline, while also improving FID. On CelebA-HQ, it leads on both attribute-combination tasks; LGD achieves a better log KID but at a substantial cost in accuracy. On ImageNet, Spectral Guidance reaches 41.6% accuracy (vs. 40.9% for TFG).

For CLIP guidance, Spectral Guidance attains the best VQAScore, while the training-free baselines drop markedly relative to their categorical performance. Fig. 3 shows clearer attribute realization and fewer off-target generations

*Table 1.* **Computational complexity comparison of guidance approaches.** $\epsilon_\theta$ and $f_\phi$ represent forward passes of the denoiser and the spectral network, respectively. $N_{\text{rec}}$ denotes the number of recursive steps required by UGD, FreeDoM and TFG.

| Method | Offline preprocessing | Weights | Per-step sampling cost | Backpropagation required |
|---|---|---|---|---|
| CG | Train Classifier | $M_{\text{cls}}$ | $\mathcal{O}(\epsilon_\theta + f_{\text{cls}} + \nabla_x f_{\text{cls}})$ | Grad through classifier |
| TFG | None | $M_{\text{cls}}$ | $\mathcal{O}(N_{\text{rec}}(\epsilon_\theta + f_{\text{cls}} + \nabla_x \epsilon_\theta + \nabla_x f_{\text{cls}}))$ | Grad through denoiser and classifier |
| **Ours** | Train $f_\phi$ + Compute $\{\Phi_t\}_\mathcal{T}$ | $M_\phi$ | $\mathcal{O}(\epsilon_\theta + f_\phi + \nabla_x f_\phi)$ | Grad through $f_\phi$ only |

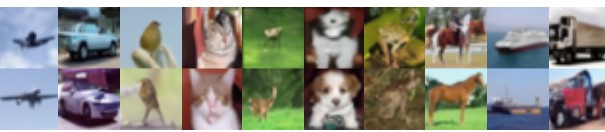

(a) Spectral Guidance (Ours)

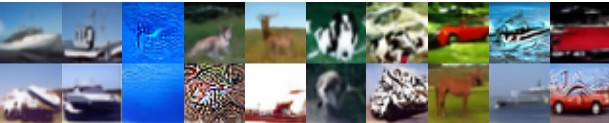

(b) Training-Free Guidance (TFG)

*Figure 2.* **Qualitative comparison on CIFAR-10.** Spectral Guidance (a) generates high-fidelity samples with clear class semantics.

than DPS, particularly for prompts involving localized attributes (sunglasses, beards).

For mask guidance, Spectral Guidance reaches an IoU of 0.80 (vs. 0.78 for TFG and FreeDoM), reusing the same $\{\Phi_t\}_{t\in\mathcal{T}}$ learned for the categorical and CLIP tasks. Qualitative examples in Fig. 4 show high-fidelity faces generated by Spectral Guidance, with hairlines aligning with the target masks (red boundary).

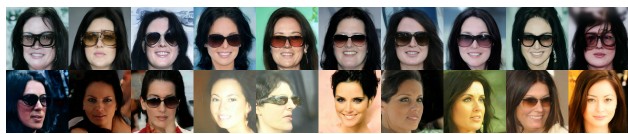

(a) Prompt: *a young woman wearing sunglasses*

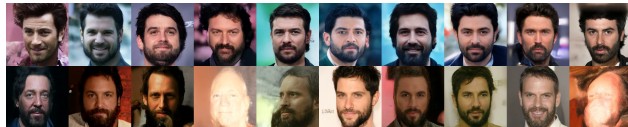

(b) Prompt: *a dark haired man with a beard*

*Figure 3.* **Qualitative comparison of CLIP guidance.** Top rows: Spectral Guidance (ours); Bottom rows: DPS.

### 6.3. Rank and Guidance Strength Ablations

**Accuracy vs. FID.** In Fig. 5(a), we analyze the trade-off between guidance strength $\kappa$ and sample quality. Spectral Guidance achieves significantly higher accuracy at equivalent FID levels. While supervised Classifier Guidance (CG) yields better accuracy for the same FID, it requires training a dedicated noise-aware classifier for every new task. In con-

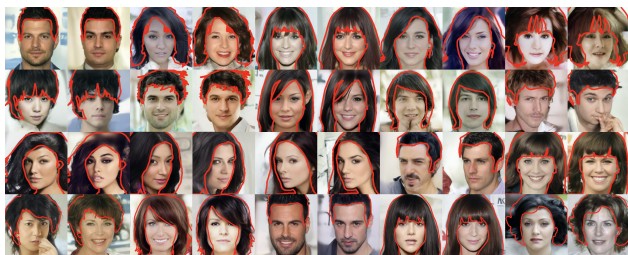

*Figure 4.* **Mask guidance.** Images generated with Spectral Guidance and target hair masks, indicated by the red boundary.

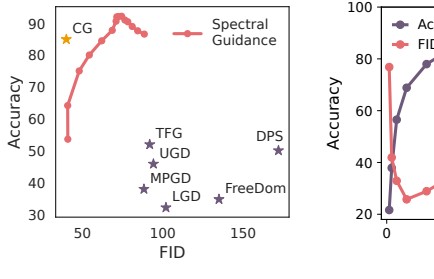
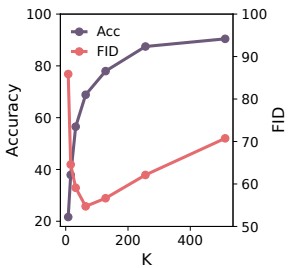

(a) Accuracy vs. FID  (b) Ablation of Rank $K$

*Figure 5.* **CIFAR-10 analysis.** (a) Sweeping guidance strength $\kappa$, Spectral Guidance achieves a better accuracy-FID frontier than training-free baselines. (b) Sweeping rank $K$ at fixed $\kappa$: accuracy improves with diminishing returns, while FID is non-monotonic, mirroring the fidelity-diversity trade-off in (a).

trast, Spectral Guidance approaches this performance using a single, unsupervised representation. At extreme guidance scales (rightmost points), we observe a degradation of FID, as guidance dominates the unconditional score, driving the sampling trajectory off the data manifold.

**Sensitivity to rank ($K$).** Fig. 5(b) ablates the dimension of the spectral approximation. Accuracy improves monotonically in $K$, rising sharply between 8 and 128 and saturating thereafter. This behavior is predicted by Proposition A.11, which bounds the $L^2(p_t)$ error of the rank-$K$ approximation of $\mathbb{E}[h(X_0) \mid x_t]$: the rapid decay of the singular spectrum (§4) ensures the leading modes capture the bulk of the recoverable class information. Beyond this regime, $K$ takes on the role of a guidance-strength knob, with each added mode raising the effective scale at fixed guidance strength $\kappa$. This sharpens class typicality, further improving accuracy at the cost of within-class diversity, mirroring the well-known

*Table 2.* **Guidance evaluation.** Validity is measured via accuracy, IoU, or VQAScore (↑); fidelity is measured via FID or log KID (↓). NG = No guidance (unconditional sampling).

| Dataset / Task | Metric | NG | DPS | LGD | FreeDoM | MPGD | UGD | TFG | **Ours** |
|---|---|---|---|---|---|---|---|---|---|
| CIFAR-10 / Labels | Accuracy (↑) | 10.0 | 50.1 | 32.2 | 34.8 | 38.0 | 45.9 | 52.0 | **89.4** |
| | FID (↓) | 98.1 | 172 | 102 | 135 | 88.3 | 94.2 | 91.7 | **70.7** |
| CelebA-HQ / Gender + Age | Accuracy (↑) | 25.0 | 71.0 | 52.0 | 68.7 | 68.6 | 75.1 | 75.2 | **91.5** |
| | log KID (↓) | -3.22 | -4.26 | **-5.10** | -3.89 | -4.79 | -4.37 | -3.86 | -2.98 |
| CelebA-HQ / Gender + Hair | Accuracy (↑) | 22.4 | 73.0 | 55.0 | 67.1 | 63.9 | 71.3 | 76.0 | **88.3** |
| | log KID (↓) | -3.13 | -3.90 | **-5.00** | -3.50 | -4.33 | -4.12 | -3.60 | -3.34 |
| ImageNet / Labels | Accuracy (↑) | 0.0 | 38.8 | 11.5 | 19.7 | 6.8 | 25.5 | 40.9 | **41.6** |
| | FID (↓) | 209 | 193 | 210 | 200 | 239 | 205 | **176** | 183 |
| CelebA-HQ / Mask | IoU (↑) | 0.38 | 0.75 | 0.45 | 0.78 | 0.48 | 0.69 | 0.78 | **0.80** |
| | log KID (↓) | **-4.62** | -3.17 | -4.08 | -3.00 | -4.50 | -3.62 | -3.15 | -2.96 |
| CelebA-HQ / CLIP | VQAScore (↑) | 0.34 | 0.59 | 0.50 | 0.57 | 0.40 | 0.44 | 0.62 | **0.64** |

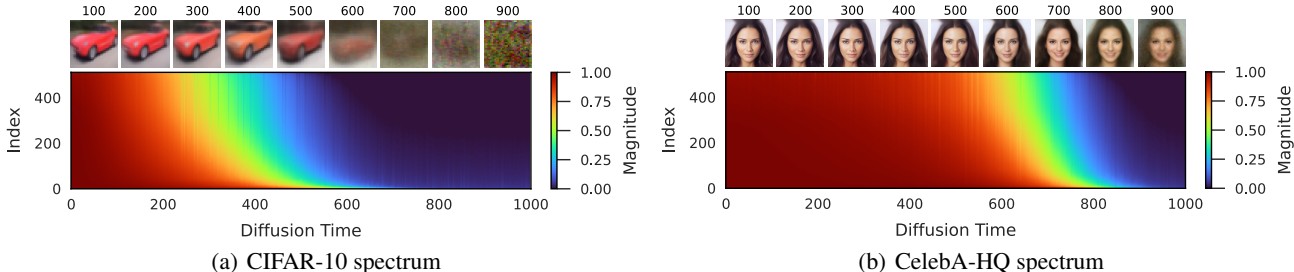

(a) CIFAR-10 spectrum        (b) CelebA-HQ spectrum

*Figure 6.* **Spectrum of the covariance operator $T_t T_t^*$.** We visualize the eigenvalues of the covariance operator $T_t T_t^*$ over diffusion time $t \in [0, 1000]$, sorted by index. The top row displays the corresponding posterior mean reconstruction $\hat{x}_0 = \mathbb{E}[X_0 \mid x_t]$.

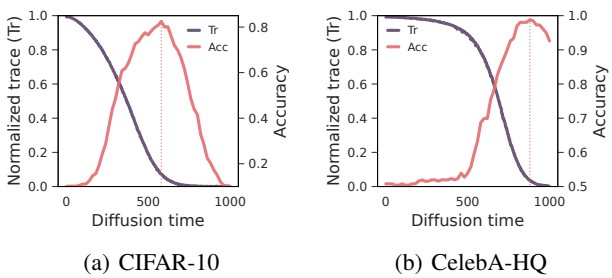

(a) CIFAR-10        (b) CelebA-HQ

*Figure 7.* **Guidance windows**. Guidance is most effective during the phase transition of the spectrum of $T_t T_t^*$.

For large $t$, the singular values concentrate near zero, as the forward process has erased all information. Between these two extremes we observe a time interval ($t \approx 400$ for CIFAR-10, $t \approx 700$ for CelebA-HQ) in which the spectrum transitions, coinciding with the formation of semantically meaningful posterior. This provides a possible explanation for the lower accuracy of posterior-mean-based guidance on CIFAR-10 (Table 2): the operator rank collapses midway through the forward process, leaving no stable guidance signal during early reverse steps. In contrast, CelebA-HQ retains information until higher noise levels.

To empirically validate this interpretation, we perform a sliding window experiment to identify the time interval when guidance is most effective. For a given $\tau$, we generate images while applying guidance only when the timestep $t$ falls within $[\tau - 100, \tau + 100]$. Fig. 7 plots the accuracy as a function of $\tau$. In the same plot, we overlay the normalized trace of the truncated covariance operator, which equals the mean of the covariance eigenvalues and serves as a proxy for the amount of information retained by the forward process. We observe that guidance effectiveness correlates with the spectral transition. The operator's spectrum serves thus as a reliable indicator for the guidance window: the period where the diffusion process is sufficiently flexible to be guided but sufficiently structured to retain semantic information.

fidelity-diversity trade-off of classifier-free guidance and lifting intra-class FID. Despite the worsening FID, reflecting lower class diversity, samples remain high-fidelity at $K = 512$, as shown in Fig. 10 (Appendix C).

### 6.4. Diffusion Spectrum and Guidance Window

As shown in Fig. 6, the spectra of CIFAR-10 and CelebA-HQ undergo a transition over time, separating three distinct regimes. For small $t$, the singular values of $T_t$ are close to 1. The operator is close to the identity, indicating that the forward process is nearly information-preserving. In this regime, the posterior mean resembles a clean image.

*Table 3.* **Computational cost on CelebA-HQ**. NG: No guidance. Batch size 1, DDIM 100 steps.

| Metric | NG | TFG | **Ours** |
|---|---|---|---|
| *Offline stage (fixed cost)* | | | |
| Train $f_\phi$ (GPU h) | – | – | 10.0 |
| Precompute $\{\Phi_t\}$ (GPU h) | – | – | 0.8 |
| *Online stage (variable cost)* | | | |
| Latency (ms/step) | 19.2 | 81.2 | **21.7** |
| Throughput (s/image) | 1.9 | 8.1 | **2.2** |
| Peak memory (GB) | 1.1 | 2.8 | 3.6 |
| *Effective time (10k images)* | | | |
| Includes Offline + Online (h) | 5.3 | 22.5 | **16.9** |

## 6.5. Sampling Efficiency

A key advantage of Spectral Guidance is its amortized cost. Unlike training-free methods that require backpropagation through the denoiser during sampling, our method moves the bulk of the computations offline. As shown in Table 3, while we incur an initial cost of $\sim$10 GPU hours to train $f_\phi$ on CelebA-HQ, this cost is amortized over future generations. In the sampling phase, our method requires only the gradients of $f_\phi$ (16M parameters), resulting in a per-step latency of 21.7 ms, comparable to unconditional sampling (19.2 ms) and nearly $4\times$ faster than TFG with $N_{\text{rec}} = 1$ (81.2 ms). The "Effective time" row demonstrates this trade-off for 10,000 images. Even including the one-time training cost, Spectral Guidance is more efficient than TFG.

## 7. Limitations

**Scale and latent diffusion.** Our experiments focus on pixel-space diffusion models at moderate scale; we do not directly evaluate Spectral Guidance on latent diffusion models or large-scale text-to-image foundation models. The framework, however, makes no assumption that $x_0$ lives in pixel space: the operator $T_t$ and its spectral decomposition are defined relative to the diffusion process itself. Learning $f_\phi$ in the latent space of an autoencoder and applying Spectral Guidance to a latent diffusion model is therefore a straightforward extension.

**Reference data for coefficient estimation.** Estimating $\hat{\mathbf{c}}_t$ in Eq. (23) requires evaluating both $\Phi_t$ and the guidance signal $h(x_0)$ on a reference set $\mathcal{D}_{\text{ref}}$ drawn from $p_0$, whereas training-free baselines rely on off-the-shelf losses or pretrained models. This can be a practical constraint when training data is unavailable or labeled examples are scarce. Two observations mitigate it. First, $\mathcal{D}_{\text{ref}}$ need not be large: $\hat{\mathbf{c}}_t$ estimates only $K + 1$ coefficients, and the cached $\Phi_t$ is reused across downstream tasks (Table 2). Second, when only the diffusion model is available, $\mathcal{D}_{\text{ref}}$ can in principle be drawn from the unconditional model itself.

**Pixel-level inverse problems.** For a linear inverse problem $y = Ax_0 + \eta$, conditional sampling requires the measurement constraint $Ax_0 \approx y$ to be satisfied at the pixel level. Our method restricts guidance to the span of the top-$K$ singular functions of $T_t$, a $K$-dimensional subspace of $\mathcal{H}_t$. For semantic signals such as labels, attributes, CLIP embeddings, and coarse masks, this low-rank structure is precisely what enables stable, denoiser-gradient-free control. For inverse problems, however, the constraint count $C$ typically far exceeds $K$: inpainting half a $256 \times 256 \times 3$ image imposes $C = 98{,}304$ independent constraints, leaving the $K$-dimensional subspace insufficient to drive $\|y - Ax_0\|^2$ to zero. Thus, we view Spectral Guidance as complementary to posterior mean-based guidance methods such as DPS.

## 8. Conclusion

We introduced Spectral Guidance, a framework that reframes conditional generation as a projection onto the intrinsic coordinates of the unconditional diffusion process. Because this basis is task-agnostic, our method provides a flexible and efficient guidance mechanism that eliminates both the need for expensive per-step denoiser backpropagation used in training-free guidance, and the task-specific retraining required for classifier guidance. Empirically, we demonstrate significant improvements in sampling efficiency and in controllability, from classes and attributes to text prompts and masks, over existing baselines. Beyond performance, our approach provides new insights into the internal structure of diffusion models. By learning the spectral decomposition of the diffusion operator, we make the evolution of information over time explicit, which allows us to determine when control is most effective.

## Acknowledgments

This work is funded by LARSyS FCT funding `UID/50009/2025` (DOI: `10.54499/UID/50009/2025`) and `LA/P/0083/2020` (DOI: `10.54499/LA/P/0083/2020`). G. Moreira is also supported via grant `SFRH/BD/151466/2021` through the Carnegie Mellon Portugal program. J. P. Costeira and M. Marques are also supported by the PT Smart Retail project (PRR - `02/C05-i11/2024.C645440011-00000062`), through IAPMEI - Agência para a Competitividade e Inovação.

## Impact Statement

This paper presents work whose goal is to advance the field of Machine Learning. There are many potential societal consequences of our work, none which we feel must be specifically highlighted here.

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

# A. Theoretical Results

We denote by $\mu_0$ and $\mu_t$ the measures with densities $p_0$ and the marginal $p_t(x_t) = \int p_t(x_t \mid x_0) \, d\mu_0(x_0)$, respectively, with respect to the Lebesgue measure. We consider two Hilbert spaces. The clean data space $\mathcal{H}_0 = L^2(\mu_0)$, with inner product

$$\langle f, g \rangle_{\mu_0} = \mathbb{E}_{X_0 \sim p_0}[f(X_0) \, g(X_0)], \tag{25}$$

and the noisy data space $\mathcal{H}_t = L^2(\mu_t)$, with inner product

$$\langle f, g \rangle_{\mu_t} = \mathbb{E}_{X_t \sim p_t}[f(X_t) \, g(X_t)]. \tag{26}$$

This appendix is structured as follows.

- **Appendix A.1** defines the backward operator $T_t$ and diffusion operator $T_t^*$, establishes the adjointness relation $\langle g, T_t f \rangle_{\mu_t} = \langle T_t^* g, f \rangle_{\mu_0}$ (Proposition A.3) and derives the covariance operator $T_t T_t^*$.

- **Appendix A.2** shows that $T_t$ is Hilbert–Schmidt (and hence compact) under the compact-support assumption on $p_0$ (Theorem A.4).

- **Appendix A.3** develops the singular value decomposition of $T_t$: the leading singular pair is the constant function $\psi_{t,1} \equiv 1$, $\phi_{t,1} \equiv 1$ with $\sigma_{t,1} = 1$ (Proposition A.5); $T_t$ is a contraction (Proposition A.6); all non-trivial singular values vanish as $\bar{\alpha}_t \to 0$ (Proposition A.7); and establishes a variational characterization of the leading eigenspace (Theorem A.9).

- **Appendix A.4** derives the spectral expansion of the guidance signal (Proposition A.10) and bounds the $L^2$ error of the rank-$K$ truncation (Proposition A.11).

- **Appendix A.5** provides a finite-sample error bound (Theorem A.12) for the whitened cross-correlation loss used to learn the singular functions in §4.3.

- **Appendix A.6** derives closed-form eigenfunctions for tractable priors: Gaussian (Proposition A.13) and circle (Proposition A.14).

## A.1. Conditional Operators

**Definition A.1** (Backward operator). The backward operator $T_t : \mathcal{H}_0 \to \mathcal{H}_t$ is the conditional expectation of a function of the clean data $f(x_0)$ given a noisy observation $x_t$. For $f \in \mathcal{H}_0$,

$$(T_t f)(x_t) := \mathbb{E}_{X_0 \sim p_t(\cdot \mid x_t)}[f(X_0)]. \tag{27}$$

This operator formalizes the denoising: it retains components of $f$ that are inferable from $X_t$.

**Definition A.2** (Diffusion operator). The diffusion operator $T_t^* : \mathcal{H}_t \to \mathcal{H}_0$ is the conditional expectation of $g(x_t)$ given an initial state $x_0$. For $g \in \mathcal{H}_t$,

$$(T_t^* g)(x_0) := \mathbb{E}_{X_t \sim p_t(\cdot \mid x_0)}[g(X_t)]. \tag{28}$$

The operator $T_t^*$ describes the evolution of $g$ under the corruption process.

**Proposition A.3** (Adjointness). *The operators $T_t$ and $T_t^*$ are Hilbert adjoints. For all $f \in \mathcal{H}_0$ and for all $g \in \mathcal{H}_t$ we have*

$$\langle g, T_t f \rangle_{\mu_t} = \langle T_t^* g, f \rangle_{\mu_0}, \tag{29}$$

*and $T_t T_t^* : \mathcal{H}_t \to \mathcal{H}_t$ is self-adjoint.*

*Proof.* By definition, we have

$$\langle g, T_t f \rangle_{\mu_t} = \int g(x_t) \left( \int f(x_0) p_t(x_0 \mid x_t) dx_0 \right) d\mu_t(x_t)$$

$$= \iint g(x_t) f(x_0) p_t(x_0 \mid x_t) p_t(x_t) \, dx_0 \, dx_t$$

$$= \iint g(x_t) f(x_0) p_t(x_0, x_t) \, dx_0 \, dx_t. \tag{30}$$

Similarly,

$$\langle T_t^* g, f \rangle_{\mu_0} = \int \left( \int g(x_t) p(x_t \mid x_0) dx_t \right) f(x_0) \, d\mu_0(x_0)$$

$$= \iint g(x_t) f(x_0) p_t(x_t \mid x_0) p_0(x_0) \, dx_t \, dx_0$$

$$= \iint g(x_t) f(x_0) p_t(x_0, x_t) \, dx_0 \, dx_t. \tag{31}$$

From Eqs. (30) and (31), we have that $\langle T_t^* g, f \rangle_{\mu_0} = \langle g, T_t f \rangle_{\mu_t}$. It follows that $T_t^*$ and $T_t$ are Hilbert adjoints. Since

$$\langle h, T_t T_t^* g \rangle_{\mu_t} = \langle T_t^* h, T_t^* g \rangle_{\mu_0} = \langle T_t T_t^* h, g \rangle_{\mu_t}, \tag{32}$$

the operator $T_t T_t^*$ is self-adjoint in $\mathcal{H}_t$. $\qquad\square$

**Covariance operator.**  The covariance operator $T_t T_t^*$ acts on test functions $f \in \mathcal{H}_t$. By expanding the operator definitions, we have

$$(T_t T_t^* f)(\tilde{x}_t) = \mathbb{E}_{X_0 \sim p(\cdot \mid \tilde{x}_t)}[\mathbb{E}_{X_t \sim p(\cdot \mid X_0)}[f(X_t)]]$$

$$= \iint f(x_t) p_t(x_0 \mid \tilde{x}_t) p_t(x_t \mid x_0) \, dx_t \, dx_0$$

$$= \int f(x_t) \left( \int p_t(x_0 \mid \tilde{x}_t) p_t(x_t \mid x_0) \, dx_0 \right) dx_t. \tag{33}$$

To express this as an integral operator with respect to the measure $\mu_t$, we multiply and divide by the marginal density $p_t(x_t)$

$$(T_t T_t^* f)(\tilde{x}_t) = \int f(x_t) \left( \int \frac{p_t(x_0 \mid \tilde{x}_t) p_t(x_t \mid x_0)}{p_t(x_t)} \, dx_0 \right) p_t(x_t) \, dx_t$$

$$= \int f(x_t) \left( \int \frac{p_t(x_0 \mid \tilde{x}_t) p_t(x_t \mid x_0)}{p_t(x_t)} \, dx_0 \right) d\mu_t(x_t). \tag{34}$$

Applying Bayes' rule, we substitute $\frac{p_t(x_t \mid x_0)}{p_t(x_t)} = \frac{p_t(x_0 \mid x_t)}{p_0(x_0)}$ to obtain

$$(T_t T_t^* f)(\tilde{x}_t) = \int f(x_t) \underbrace{\left( \int \frac{p_t(x_0 \mid \tilde{x}_t) p_t(x_0 \mid x_t)}{p_0(x_0)} \, dx_0 \right)}_{=: \, k_t(x_t, \tilde{x}_t)} d\mu_t(x_t). \tag{35}$$

Hence, $T_t T_t^*$ admits a symmetric diffusion kernel $k_t$ on the noisy data manifold. Defined with respect to $\mu_t$, the kernel is:

$$k_t(x_t, \tilde{x}_t) = \int \frac{p_t(x_0 \mid x_t) p_t(x_0 \mid \tilde{x}_t)}{p_0(x_0)} \, dx_0. \tag{36}$$

This kernel $k_t(x_t, \tilde{x}_t)$ represents the transition density of the coupled backward-forward process $(x_t \to x_0 \to \tilde{x}_t)$. It measures the connectivity between two noisy points $x_t$ and $\tilde{x}_t$. A large $t$ allows the kernel to capture the coarser, global structure of the data (Coifman & Lafon, 2006). From Eq. (36), $k_t$ is symmetric *i.e.*, $k_t(x_t, \tilde{x}_t) = k_t(\tilde{x}_t, x_t)$.

### A.2. Compactness

**Theorem A.4** (Compactness). *Let $p_0$ have compact support $\mathcal{X} := \mathrm{supp}(p_0)$ with $p_0 > 0$ on $\mathcal{X}$, and let the forward kernel be Gaussian, $p_t(x_t \mid x_0) = \mathcal{N}(x_t; \sqrt{\bar{\alpha}_t}\, x_0, (1 - \bar{\alpha}_t)I)$. Then, for every $t > 0$, the backward operator $T_t$ is Hilbert–Schmidt and thus, compact.*

*Proof.* We write $T_t$ in integral-kernel form. Since $d\mu_0 = p_0\, dx_0$, we have

$$(T_t f)(x_t) = \int_{\mathcal{X}} f(x_0)\, p_t(x_0 \mid x_t)\, dx_0 = \int_{\mathcal{X}} f(x_0)\, \underbrace{\frac{p_t(x_0 \mid x_t)}{p_0(x_0)}}_{=:\, k_t(x_t, x_0)}\, d\mu_0(x_0). \tag{37}$$

Because $T_t : L^2(\mu_0) \to L^2(\mu_t)$ is an integral operator with kernel $k_t$, its Hilbert–Schmidt norm satisfies

$$\|T_t\|_{\mathrm{HS}}^2 = \iint |k_t(x_t, x_0)|^2\, d\mu_0(x_0)\, d\mu_t(x_t). \tag{38}$$

Substituting $k_t = p_t(x_0 \mid x_t)/p_0(x_0)$ and applying Bayes' rule, $p_t(x_0 \mid x_t) = p_t(x_t \mid x_0)\, p_0(x_0)/p_t(x_t)$, yields

$$
\begin{aligned}
\|T_t\|_{\mathrm{HS}}^2 &= \iint \frac{p_t(x_0 \mid x_t)^2}{p_0(x_0)^2}\, d\mu_0(x_0)\, d\mu_t(x_t) \\
&= \iint \frac{p_t(x_t \mid x_0)^2}{p_t(x_t)^2}\, p_0(x_0)\, p_t(x_t)\, dx_0\, dx_t \\
&= \int_{\mathcal{X}} p_0(x_0) \left( \int_{\mathbb{R}^d} \frac{p_t(x_t \mid x_0)^2}{p_t(x_t)}\, dx_t \right) dx_0.
\end{aligned}
\tag{39}
$$

It therefore suffices to show that the inner integral is uniformly bounded for $x_0 \in \mathcal{X}$.

**Lower bound on $p_t(x_t)$.** Since $\mathcal{X}$ is compact, there exists $R > 0$ such that $\|x_0'\| \leq R$ for all $x_0' \in \mathcal{X}$. By the triangle inequality, $\|x_t - \sqrt{\bar{\alpha}_t}\, x_0'\| \leq \|x_t\| + \sqrt{\bar{\alpha}_t}\, R$, and because the Gaussian density is decreasing in $\|x_t - \sqrt{\bar{\alpha}_t}\, x_0'\|^2$, we obtain

$$p_t(x_t \mid x_0') \geq \frac{1}{\left(2\pi(1 - \bar{\alpha}_t)\right)^{d/2}} \exp\left( -\frac{\|x_t\|^2 + 2\sqrt{\bar{\alpha}_t}\, R\, \|x_t\| + \bar{\alpha}_t R^2}{2(1 - \bar{\alpha}_t)} \right). \tag{40}$$

The right-hand side is independent of $x_0'$, so marginalizing gives the same lower bound for

$$p_t(x_t) = \int_{\mathcal{X}} p_t(x_t \mid x_0')\, p_0(x_0')\, dx_0' \geq \frac{1}{\left(2\pi(1 - \bar{\alpha}_t)\right)^{d/2}} \exp\left( -\frac{\|x_t\|^2 + 2\sqrt{\bar{\alpha}_t}\, R\, \|x_t\| + \bar{\alpha}_t R^2}{2(1 - \bar{\alpha}_t)} \right). \tag{41}$$

**Bounding the ratio.** The squared numerator is

$$p_t(x_t \mid x_0)^2 = \frac{1}{\left(2\pi(1 - \bar{\alpha}_t)\right)^d} \exp\left( -\frac{\|x_t - \sqrt{\bar{\alpha}_t}\, x_0\|^2}{1 - \bar{\alpha}_t} \right). \tag{42}$$

Expanding $\|x_t - \sqrt{\bar{\alpha}_t}\, x_0\|^2 = \|x_t\|^2 - 2\sqrt{\bar{\alpha}_t}\, x_t^\top x_0 + \bar{\alpha}_t \|x_0\|^2$ and combining with the lower bound (41), the Gaussian prefactors simplify and the exponent becomes

$$\frac{-\|x_t\|^2 + 4\sqrt{\bar{\alpha}_t}\, x_t^\top x_0 + 2\sqrt{\bar{\alpha}_t}\, R\, \|x_t\| - 2\bar{\alpha}_t \|x_0\|^2 + \bar{\alpha}_t R^2}{2(1 - \bar{\alpha}_t)}. \tag{43}$$

By Cauchy–Schwarz, $x_t^\top x_0 \leq \|x_t\|\, \|x_0\| \leq R\, \|x_t\|$ for $x_0 \in \mathcal{X}$, and $-2\bar{\alpha}_t \|x_0\|^2 \leq 0$, so

$$\frac{p_t(x_t \mid x_0)^2}{p_t(x_t)} \leq \frac{1}{\left(2\pi(1 - \bar{\alpha}_t)\right)^{d/2}} \exp\left( \frac{-\|x_t\|^2 + 6\sqrt{\bar{\alpha}_t}\, R\, \|x_t\| + \bar{\alpha}_t R^2}{2(1 - \bar{\alpha}_t)} \right). \tag{44}$$

**Completing the square.** Writing $u = \|x_t\|$ and $c = 3\sqrt{\bar{\alpha}_t}\,R$, we have

$$-u^2 + 6\sqrt{\bar{\alpha}_t}\,R\,u + \bar{\alpha}_t R^2 = -(u-c)^2 + 9\bar{\alpha}_t R^2 + \bar{\alpha}_t R^2 = -(u-c)^2 + 10\bar{\alpha}_t R^2. \tag{45}$$

To make the Gaussian integral tractable, we apply the bound $(u-c)^2 \geq \frac{1}{2}u^2 - c^2$, which follows from Young's inequality with $\epsilon = 1/2$. This gives

$$-(u-c)^2 + 10\bar{\alpha}_t R^2 \leq -\tfrac{1}{2}\|x_t\|^2 + 9\bar{\alpha}_t R^2 + 10\bar{\alpha}_t R^2 = -\tfrac{1}{2}\|x_t\|^2 + 19\bar{\alpha}_t R^2. \tag{46}$$

Substituting into (44),

$$\frac{p_t(x_t \mid x_0)^2}{p_t(x_t)} \leq \frac{1}{\left(2\pi(1-\bar{\alpha}_t)\right)^{d/2}}\, \exp\left(\frac{19\bar{\alpha}_t R^2}{2(1-\bar{\alpha}_t)}\right)\, \exp\left(-\frac{\|x_t\|^2}{4(1-\bar{\alpha}_t)}\right). \tag{47}$$

**Concluding the bound.** Integrating (47) over $x_t \in \mathbb{R}^d$ yields the Gaussian integral

$$\int_{\mathbb{R}^d} \frac{p_t(x_t \mid x_0)^2}{p_t(x_t)}\,dx_t \leq \frac{\left(4\pi(1-\bar{\alpha}_t)\right)^{d/2}}{\left(2\pi(1-\bar{\alpha}_t)\right)^{d/2}}\, \exp\left(\frac{19\bar{\alpha}_t R^2}{2(1-\bar{\alpha}_t)}\right) = 2^{d/2}\,\exp\left(\frac{19\bar{\alpha}_t R^2}{2(1-\bar{\alpha}_t)}\right) =: C(t, R). \tag{48}$$

The constant $C(t, R)$ is finite for every $t > 0$ and depends only on $t$, $R$, and the dimension $d$. Substituting back into (39),

$$\|T_t\|_{\mathrm{HS}}^2 \leq C(t, R) \int_{\mathcal{X}} p_0(x_0)\,dx_0 = C(t, R) < \infty. \tag{49}$$

Since $T_t$ has finite Hilbert–Schmidt norm, it is Hilbert–Schmidt and therefore compact. $\qquad\square$

By proving compactness for $T_t$, we immediately have that the adjoint $T_t^*$ is compact. Consequently, the covariance operator $T_t T_t^*$, being the composition of compact operators, is itself compact.

### A.3. Singular Value Decomposition

Since $T_t$ is compact (Theorem A.4), it admits a singular value decomposition (SVD). For any $f \in L^2(\mu_0)$,

$$(T_t f)(x_t) = \sum_{k=1}^{\infty} \sigma_{t,k}\,\phi_{t,k}(x_t)\,\langle \psi_{t,k}, f\rangle_{\mu_0}, \tag{50}$$

where the convergence is in $L^2(\mu_t)$, $\{\psi_{t,k}\}_{k\geq 1} \subset L^2(\mu_0)$ and $\{\phi_{t,k}\}_{k\geq 1} \subset L^2(\mu_t)$ are orthonormal systems, and $\sigma_{t,1} \geq \sigma_{t,2} \geq \cdots \geq 0$ are the singular values. In particular,

$$T_t^* T_t\,\psi_{t,k} = \sigma_{t,k}^2\,\psi_{t,k}, \qquad T_t^*\,\phi_{t,k} = \sigma_{t,k}\,\psi_{t,k}. \tag{51}$$

**Proposition A.5** (Leading singular functions). *Let $\psi_{t,1} \equiv 1$ and $\phi_{t,1} \equiv 1$ denote the constant functions equal to one. Then $(\psi_{t,1}, \phi_{t,1})$ is a singular pair of $T_t$ with singular value $\sigma_{t,1} = 1$. That is,*

$$T_t\,\psi_{t,1} = \phi_{t,1}, \qquad T_t^*\,\phi_{t,1} = \psi_{t,1}. \tag{52}$$

*Proof.* We verify three facts: that $T_t$ and $T_t^*$ each map $\mathbf{1}$ to $\mathbf{1}$, and that the singular functions have unit norm.

For every $x_t$,

$$(T_t\,\mathbf{1})(x_t) = \mathbb{E}[\mathbf{1}(X_0) \mid X_t = x_t] = \int_{\mathcal{X}} p_t(x_0 \mid x_t)\,dx_0 = 1, \tag{53}$$

since $p(\cdot \mid x_t)$ is a probability density over $x_0$. Hence $T_t\,\mathbf{1} = \mathbf{1}$.

Similarly, for every $x_0$,

$$(T_t^*\,\mathbf{1})(x_0) = \mathbb{E}[\mathbf{1}(X_t) \mid X_0 = x_0] = \int_{\mathbb{R}^d} p_t(x_t \mid x_0)\,dx_t = 1, \tag{54}$$

since $p(\cdot \mid x_0)$ is a probability density over $x_t$. Hence $T_t^* \mathbf{1} = \mathbf{1}$.

The singular functions have unit norm in their respective spaces:

$$\|\psi_{t,1}\|_{\mu_0}^2 = \mathbb{E}_{X_0 \sim p_0}[\mathbf{1}^2] = 1, \qquad \|\phi_{t,1}\|_{\mu_t}^2 = \mathbb{E}_{X_t \sim p_t}[\mathbf{1}^2] = 1. \tag{55}$$

Combining the above, $T_t \psi_{t,1} = \phi_{t,1}$ and $T_t^* \phi_{t,1} = \psi_{t,1}$, so $(\psi_{t,1}, \phi_{t,1})$ is a singular pair with singular value $\sigma_{t,1} = 1$. $\quad\square$

**Proposition A.6** ($T_t$ is a contraction). *For every $f \in L^2(\mu_0)$, $\|T_t f\|_{\mu_t} \leq \|f\|_{\mu_0}$. In particular, $\sigma_{t,k} \leq 1$ for all $k \geq 1$.*

*Proof.* Fix $f \in L^2(\mu_0)$. Applying Jensen's inequality to the conditional expectation and then the tower property gives

$$\begin{aligned}
\|T_t f\|_{\mu_t}^2 &= \mathbb{E}_{X_t \sim p_t}\big[\mathbb{E}[f(X_0) \mid X_t]^2\big] \\
&\leq \mathbb{E}_{X_t \sim p_t}\big[\mathbb{E}[f(X_0)^2 \mid X_t]\big] &\text{(Jensen)} \\
&= \mathbb{E}_{X_0 \sim p_0}[f(X_0)^2] &\text{(tower property)} \\
&= \|f\|_{\mu_0}^2.
\end{aligned}$$

Hence $\|T_t\|_{\mathrm{op}} \leq 1$, and since $\|T_t\|_{\mathrm{op}} = \sigma_{t,1} \geq \sigma_{t,2} \geq \cdots$, it follows that $\sigma_{t,k} \leq 1$ for every $k \geq 1$. $\quad\square$

**Proposition A.7** (Non-trivial singular values vanish as $\bar{\alpha}_t \to 0$). *Let $p_0$ have compact support $\mathcal{X}$ and let $X_t = \sqrt{\bar{\alpha}_t}\, X_0 + \sqrt{1 - \bar{\alpha}_t}\, \epsilon$, with $\epsilon \sim \mathcal{N}(0, I)$. Then, for every $k \geq 2$,*

$$\sigma_{t,k} \to 0 \quad as \quad \bar{\alpha}_t \to 0. \tag{56}$$

*Proof.* Since $\sigma_{t,1} = 1$ with singular functions $\psi_{t,1} \equiv 1$, $\phi_{t,1} \equiv 1$ (Proposition A.5), it suffices to show that the operator norm of $T_t$ restricted to $\mathbf{1}^\perp := \{f \in L^2(\mu_0) : \mathbb{E}_{p_0}[f] = 0\}$ vanishes as $\bar{\alpha}_t \to 0$.

**Rewriting the operator on $\mathbf{1}^\perp$.** Fix $f \in \mathbf{1}^\perp$, so that $\int f(x_0)\, p_0(x_0)\, dx_0 = 0$. Applying Bayes' rule, $p(x_0 \mid x_t) = p(x_t \mid x_0)\, p_0(x_0)/p_t(x_t)$, and using the zero-mean condition to subtract the identity,

$$\begin{aligned}
(T_t f)(x_t) &= \int_{\mathcal{X}} f(x_0)\, p_t(x_0 \mid x_t)\, dx_0 \\
&= \int_{\mathcal{X}} f(x_0)\, \frac{p_t(x_t \mid x_0)}{p_t(x_t)}\, p_0(x_0)\, dx_0 \\
&= \int_{\mathcal{X}} f(x_0) \left( \frac{p_t(x_t \mid x_0)}{p_t(x_t)} - 1 \right) p_0(x_0)\, dx_0, \tag{57}
\end{aligned}$$

where the last step uses $\int f\, d\mu_0 = 0$.

**Pointwise and integrated bound.** Applying Cauchy–Schwarz in $L^2(\mu_0)$ to (57) gives, for each $x_t$,

$$\big|(T_t f)(x_t)\big|^2 \leq \|f\|_{\mu_0}^2 \int_{\mathcal{X}} \left( \frac{p_t(x_t \mid x_0)}{p_t(x_t)} - 1 \right)^2 p_0(x_0)\, dx_0. \tag{58}$$

Integrating both sides against $\mu_t$ *i.e.*, multiplying by $p_t(x_t)$ and integrating over $x_t$,

$$\|T_t f\|_{\mu_t}^2 \leq \|f\|_{\mu_0}^2 \int_{\mathcal{X}} p_0(x_0) \int_{\mathbb{R}^d} \frac{(p_t(x_t \mid x_0) - p_t(x_t))^2}{p_t(x_t)}\, dx_t\, dx_0 = \|f\|_{\mu_0}^2\, \mathbb{E}_{X_0 \sim p_0}\big[\chi^2\big(p_t(\cdot \mid X_0) \,\|\, p_t\big)\big], \tag{59}$$

where $\chi^2(q\|r) := \int (q - r)^2/r$ denotes the chi-squared divergence.

**Vanishing of the chi-squared divergence.** As $\bar{\alpha}_t \to 0$, the forward kernel $p_t(x_t \mid x_0) = \mathcal{N}(x_t; \sqrt{\bar{\alpha}_t}\, x_0, (1 - \bar{\alpha}_t)\, I)$ converges to $\mathcal{N}(x_t; 0, I)$. Since $\mathcal{X}$ is compact, this convergence is uniform over $x_0 \in \mathcal{X}$: for every $x_t$,

$$\sup_{x_0 \in \mathcal{X}} \left| \frac{p_t(x_t \mid x_0)}{p_t(x_t)} - 1 \right| \longrightarrow 0 \quad \text{as} \quad \bar{\alpha}_t \to 0, \tag{60}$$

because both the numerator and the denominator $p_t(x_t) = \int_{\mathcal{X}} p_t(x_t \mid x_0')\, p_0(x_0')\, dx_0'$ converge to the same standard Gaussian density. Therefore,

$$\mathbb{E}_{X_0 \sim p_0}\big[\chi^2\big(p_t(\cdot \mid X_0) \,\|\, p_t\big)\big] \longrightarrow 0 \quad \text{as} \quad \bar{\alpha}_t \to 0. \tag{61}$$

**Conclusion.** Taking the supremum of (59) over $f \in \mathbf{1}^{\perp}$ with $\|f\|_{\mu_0} = 1$ gives

$$\sigma_{t,2}^2 = \big\| T_t\big|_{\mathbf{1}^{\perp}} \big\|_{\mathrm{op}}^2 \leq \mathbb{E}_{X_0 \sim p_0}\big[\chi^2\big(p_t(\cdot \mid X_0) \,\|\, p_t\big)\big] \longrightarrow 0. \tag{62}$$

Since $\sigma_{t,2} \geq \sigma_{t,3} \geq \cdots$, it follows that $\sigma_{t,k} \to 0$ for all $k \geq 2$. $\qquad\square$

We conclude with the variational characterization of the SVD of $T_t$, which forms the basis of our learning algorithm. We first present the constrained formulation, based on the Courant-Fischer theorem and then the unconstrained approach.

---

**Lemma A.8** (Constrained variational characterization of the leading eigenspace). *Let $\zeta$ denote the joint distribution of two independent noisy views of the same clean sample,*

$$\zeta(x_t, \tilde{x}_t) := \int_{\mathcal{X}} p_t(x_t \mid x_0)\, p_t(\tilde{x}_t \mid x_0)\, d\mu_0(x_0). \tag{63}$$

*Then the top-$K$ left singular functions $\{\phi_{t,k}\}_{k=1}^{K}$ of $T_t$ solve*

$$\max_{f_1, \ldots, f_K} \sum_{k=1}^{K} \mathbb{E}_{(X_t, \tilde{X}_t) \sim \zeta}\big[f_k(X_t)\, f_k(\tilde{X}_t)\big] \qquad \text{s.t.} \quad \langle f_i, f_j \rangle_{\mu_t} = \delta_{ij}. \tag{64}$$

*The maximum value is $\sum_{k=1}^{K} \lambda_{t,k}$, where $\lambda_{t,k} = \sigma_{t,k}^2$. Given any maximizer $\{\phi_{t,k}\}_{k=1}^{K}$, the corresponding right singular functions are recovered as*

$$\psi_{t,k}(x_0) = \frac{1}{\sigma_{t,k}} (T_t^* \phi_{t,k})(x_0) = \frac{1}{\sigma_{t,k}} \mathbb{E}\big[\phi_{t,k}(X_t) \mid X_0 = x_0\big]. \tag{65}$$

---

*Proof.* Since $T_t T_t^* : \mathcal{H}_t \to \mathcal{H}_t$ is compact (Theorem A.4) and self-adjoint, the Courant–Fischer theorem gives

$$\max_{\substack{f_1, \ldots, f_K \in \mathcal{H}_t \\ \langle f_i, f_j \rangle_{\mu_t} = \delta_{ij}}} \sum_{k=1}^{K} \langle f_k, T_t T_t^* f_k \rangle_{\mu_t} = \sum_{k=1}^{K} \lambda_{t,k}, \tag{66}$$

attained when $\mathrm{span}\{f_1, \ldots, f_K\} = \mathrm{span}\{\phi_{t,1}, \ldots, \phi_{t,K}\}$. It remains to show that the Rayleigh quotient equals the cross-view correlation.

**Expanding the Rayleigh quotient.** For $f \in \mathcal{H}_t$, we unwind the operator composition $T_t T_t^*$ by first expanding the outer operator $T_t$ (expectation over $X_0 \mid X_t$), then the inner operator $T_t^*$ (expectation over $\tilde{X}_t \mid X_0$):

$$\begin{aligned}
\langle f, T_t T_t^* f \rangle_{\mu_t} &= \mathbb{E}_{X_t \sim p_t}\big[f(X_t)\, (T_t T_t^* f)(X_t)\big] \\
&= \mathbb{E}_{X_t \sim p_t}\Big[f(X_t)\, \mathbb{E}\big[(T_t^* f)(X_0) \mid X_t\big]\Big] && \text{(expanding } T_t\text{)} \\
&= \mathbb{E}_{X_t \sim p_t}\Big[f(X_t)\, \mathbb{E}\big[\mathbb{E}[f(\tilde{X}_t) \mid X_0] \mid X_t\big]\Big] && \text{(expanding } T_t^*\text{)} \\
&= \mathbb{E}_{X_t, X_0, \tilde{X}_t}\big[f(X_t)\, f(\tilde{X}_t)\big] && \text{(tower property)} \\
&= \mathbb{E}_{(X_t, \tilde{X}_t) \sim \zeta}\big[f(X_t)\, f(\tilde{X}_t)\big], && (67)
\end{aligned}$$

where the last equality follows because the marginal over $(X_t, \tilde{X}_t)$, obtained by integrating out $X_0$, is precisely $\zeta$. Substituting (67) into (66) yields the variational problem (64).

**Recovery of the right singular functions.** The SVD relation $T_t^* \phi_{t,k} = \sigma_{t,k} \psi_{t,k}$ (cf. Eq. (51)) gives $\psi_{t,k} = \sigma_{t,k}^{-1} T_t^* \phi_{t,k}$, and expanding the definition of $T_t^*$ yields (65). $\qquad\square$

---

**Theorem A.9** (Variational characterization). *For any* $f = (f_1, \ldots, f_K)^\top$ *with* $\mathbb{E}_{p_t}[f_k] = 0$, *define the* $K \times K$ *covariance matrix*

$$\mathbf{\Sigma}_t(f) := \mathbb{E}_{p_t}\left[ f(X_t) f(X_t)^\top \right] \succ 0 \tag{68}$$

*and the cross-covariance*

$$\mathbf{C}_t(f) := \mathbb{E}_\zeta\left[ f(X_t) f(\tilde{X}_t)^\top \right]. \tag{69}$$

*Then*

$$\max_f \; \mathrm{Tr}\big(\mathbf{C}_t(f)\,\mathbf{\Sigma}_t(f)^{-1}\big) \;=\; \sum_{k=2}^{K+1} \sigma_{t,k}^2 \tag{70}$$

*and any maximizer* $f^\star$ *satisfies* $\mathrm{span}\{f_k^\star\}_{k=1}^K = \mathrm{span}\{\phi_{t,k}\}_{k=2}^{K+1}$.

---

*Proof.* For any valid $f$, let $g := \mathbf{\Sigma}_t(f)^{-1/2} f$. Then $\mathbb{E}_{p_t}[gg^\top] = \mathbf{I}$, so $g$ satisfies the $L^2(\mu_t)$ orthonormality constraint of Lemma A.8. Using cyclicity of the trace, a direct calculation gives

$$\mathrm{Tr}\big(\mathbf{C}_t(f)\,\mathbf{\Sigma}_t(f)^{-1}\big) = \mathrm{Tr}\Big(\mathbf{\Sigma}_t(f)^{-1/2}\mathbf{C}_t(f)\,\mathbf{\Sigma}_t(f)^{-1/2}\Big) = \mathrm{Tr}(\mathbf{C}_t(g)) = \sum_{k=1}^{K}\mathbb{E}_\zeta\Big[g_k(X_t)\,g_k(\tilde{X}_t)\Big]. \tag{71}$$

Conversely, any $g$ satisfying the orthonormality constraint is itself admissible in the unconstrained problem (with $\mathbf{\Sigma}_t(g) = \mathbf{I}$), and both objectives take the same value on such $g$. The map $f \mapsto g$ is therefore a bijection between admissible points of the two problems that preserves the objective, so they share the same supremum. By Lemma A.8, this common supremum equals $\sum_{k=2}^{K+1} \sigma_{t,k}^2$ and is attained when $\{g_k\}_{k=1}^K$ is any orthonormal basis of $\mathrm{span}\{\phi_{t,k}\}_{k=2}^{K+1}$. Since $f = \mathbf{\Sigma}_t(f)^{1/2}g$ is an invertible linear transformation, $\mathrm{span}\{f_k^\star\}_{k=1}^K = \mathrm{span}\{g_k\}_{k=1}^K = \mathrm{span}\{\phi_{t,k}\}_{k=2}^{K+1}$, as claimed. $\qquad\square$

## A.4. Spectral Guidance

---

**Proposition A.10** (Noisy posterior via the SVD). *Let* $h \in L^2(\mu_0)$. *Then*

$$\mathbb{E}_{p(\cdot|x_t)}[h(X_0)] = \sum_{k=1}^{\infty} c_{t,k}\, \phi_{t,k}(x_t), \qquad c_{t,k} := \mathbb{E}_{(X_0,X_t)\sim p(x_0,x_t)}\big[h(X_0)\,\phi_{t,k}(X_t)\big]. \tag{72}$$

---

*Proof.* Since $\mathbb{E}[h(X_0) \mid X_t = x_t] = (T_t h)(x_t)$, the SVD expansion gives

$$\mathbb{E}[h(X_0) \mid X_t = x_t] = \sum_{k=1}^{\infty} \sigma_{t,k}\, \phi_{t,k}(x_t)\, \langle h,\, \psi_{t,k}\rangle_{\mu_0}. \tag{73}$$

We rewrite the inner product using the SVD relation $\psi_{t,k} = \frac{1}{\sigma_{t,k}} T_t^* \phi_{t,k}$ :

$$\begin{aligned}
\sigma_{t,k}\, \langle h,\, \psi_{t,k}\rangle_{\mu_0} &= \sigma_{t,k} \cdot \frac{1}{\sigma_{t,k}}\, \langle h,\, T_t^*\phi_{t,k}\rangle_{\mu_0} \\
&= \mathbb{E}_{X_0\sim p_0}\big[h(X_0)\,(T_t^*\phi_{t,k})(X_0)\big] \\
&= \mathbb{E}_{X_0\sim p_0}\Big[h(X_0)\,\mathbb{E}\big[\phi_{t,k}(X_t)\mid X_0\big]\Big] \\
&= \mathbb{E}_{(X_0,X_t)\sim p(x_0,x_t)}\big[h(X_0)\,\phi_{t,k}(X_t)\big] =: c_{t,k},
\end{aligned} \tag{74}$$

where the third line expands the definition of $T_t^*$ and the fourth applies the tower property. Substituting into (73) yields the result. $\qquad\square$

We next quantify the error incurred by truncating the SVD expansion at rank $K$. Define the rank-$K$ approximation

$$(T_{t,K}h)(x_t) := \sum_{k=1}^{K} \sigma_{t,k}\, \phi_{t,k}(x_t)\, \langle h,\, \psi_{t,k} \rangle_{\mu_0}, \tag{75}$$

and write $\lambda_{t,k} := \sigma_{t,k}^2$ for the eigenvalues of $T_t T_t^*$.

> **Proposition A.11** (SVD truncation error)**.** *Let $T_{t,K}$ be the rank-K approximation of $T_t$. For any $h \in L^2(\mu_0)$,*
>
> $$\|T_t h - T_{t,K} h\|_{\mu_t}^2 \;\leq\; \lambda_{t,K+1}\, \|h\|_{\mu_0}^2. \tag{76}$$

*Proof.* By orthonormality of $\{\phi_{t,k}\}_{k \geq 1}$ in $L^2(\mu_t)$, the squared error is the tail of the expansion

$$\|T_t h - T_{t,K} h\|_{\mu_t}^2 \;=\; \sum_{k=K+1}^{\infty} \sigma_{t,k}^2\, \langle h,\, \psi_{t,k} \rangle_{\mu_0}^2. \tag{77}$$

Since the singular values are ordered, $\sigma_{t,k}^2 \leq \sigma_{t,K+1}^2 = \lambda_{t,K+1}$ for all $k \geq K+1$. Factoring out the leading term and applying Bessel's inequality,

$$\sum_{k=K+1}^{\infty} \sigma_{t,k}^2\, \langle h,\, \psi_{t,k} \rangle_{\mu_0}^2 \;\leq\; \lambda_{t,K+1} \sum_{k=K+1}^{\infty} \langle h,\, \psi_{t,k} \rangle_{\mu_0}^2 \;\leq\; \lambda_{t,K+1}\, \|h\|_{\mu_0}^2. \tag{78}$$
$\qquad\square$

## A.5. Finite Sample Error

> **Theorem A.12** (Finite-sample error of the variational objective)**.** *Let $f : \mathcal{X} \to \mathbb{R}^K$ with $\mathbb{E}_{p_t}[f(X_t)] = 0$ and $\|f(X_t)\|_2 \leq M$ almost surely. Assume the population covariance $\boldsymbol{\Sigma}_t(f) := \mathbb{E}_{p_t}\left[f(X_t)\,f(X_t)^\top\right]$ satisfies $\boldsymbol{\Sigma}_t(f) \succeq \lambda \mathbf{I}$ for some $\lambda > 0$. Given $B$ i.i.d. pairs $\left\{(X_t^{(i)}, \tilde{X}_t^{(i)})\right\}_{i=1}^{B} \sim \zeta$, define the symmetrized empirical estimators*
>
> $$\hat{\mathbf{C}}_t(f) := \frac{1}{B} \sum_{i=1}^{B} \tfrac{1}{2}\Big(f(X_t^{(i)})f(\tilde{X}_t^{(i)})^\top + f(\tilde{X}_t^{(i)})f(X_t^{(i)})^\top\Big), \qquad \hat{\boldsymbol{\Sigma}}_t(f) := \frac{1}{B} \sum_{i=1}^{B} f(X_t^{(i)})f(X_t^{(i)})^\top, \tag{79}$$
>
> *and the population and empirical objectives*
>
> $$J_t(f) := \mathrm{Tr}\big(\mathbf{C}_t(f)\,\boldsymbol{\Sigma}_t(f)^{-1}\big), \qquad \hat{J}_t(f) := \mathrm{Tr}\big(\hat{\mathbf{C}}_t(f)\,\hat{\boldsymbol{\Sigma}}_t(f)^{-1}\big). \tag{80}$$
>
> *Then for any $\delta \in (0,1)$ and any $B$ satisfying $B \geq 32\,M^4 \log(4K/\delta)/\lambda^2$, with probability at least $1 - \delta$,*
>
> $$\left|\hat{J}_t(f) - J_t(f)\right| \;\leq\; \frac{4\,K\,M^4}{\lambda^2} \sqrt{\frac{\log(4K/\delta)}{B}} \;+\; \frac{16\,K\,M^4 \log(4K/\delta)}{3\,\lambda^2\,B}. \tag{81}$$

*Proof.* Let $\mathbf{H}_i := \tfrac{1}{2}\big(f(X_t^{(i)})f(\tilde{X}_t^{(i)})^\top + f(\tilde{X}_t^{(i)})f(X_t^{(i)})^\top\big)$ and $\mathbf{S}_i := f(X_t^{(i)})f(X_t^{(i)})^\top$, both self-adjoint $K \times K$ matrices. By exchangeability of $(X_t^{(i)}, \tilde{X}_t^{(i)})$ under $\zeta$, $\mathbb{E}[\mathbf{H}_i] = \mathbf{C}_t(f)$; trivially $\mathbb{E}[\mathbf{S}_i] = \boldsymbol{\Sigma}_t(f)$. Define centered summands $\mathbf{Y}_i := \mathbf{H}_i - \mathbf{C}_t(f)$ and $\mathbf{T}_i := \mathbf{S}_i - \boldsymbol{\Sigma}_t(f)$.

Since $\|f\|_2 \leq M$ a.s., $\|\mathbf{H}_i\|, \|\mathbf{S}_i\| \leq M^2$, hence $\|\mathbf{Y}_i\|, \|\mathbf{T}_i\| \leq 2M^2$. For the variance proxies,

$$\mathbb{E}[\mathbf{Y}_i^2] = \mathbb{E}[\mathbf{H}_i^2] - \mathbf{C}_t^2 \preceq \mathbb{E}[\mathbf{H}_i^2], \tag{82}$$

since $\mathbf{C}_t^2 \succeq 0$. By Jensen, $\|\mathbb{E}[\mathbf{H}_i^2]\| \leq \mathbb{E}\big[\|\mathbf{H}_i\|^2\big] \leq M^4$, so $\|\mathbb{E}[\mathbf{Y}_i^2]\| \leq M^4$; the same bound applies to $\mathbb{E}[\mathbf{T}_i^2]$.

**Concentration of $\hat{\mathbf{C}}_t$ and $\hat{\boldsymbol{\Sigma}}_t$.** Applying matrix Bernstein (Tropp, 2015) to $\sum_i \mathbf{Y}_i$ in its two-sided form, then to $\sum_i \mathbf{T}_i$, and union-bounding over both estimators (with $\delta/2$ each), gives that with probability at least $1 - \delta$,

$$\max\big(\|\hat{\mathbf{C}}_t - \mathbf{C}_t\|, \|\hat{\boldsymbol{\Sigma}}_t - \boldsymbol{\Sigma}_t\|\big) \leq \varepsilon_B := M^2\left(\sqrt{\frac{2\log(4K/\delta)}{B}} + \frac{4\log(4K/\delta)}{3B}\right). \tag{83}$$

**Perturbation of the trace.** Write $\Delta_C := \hat{\mathbf{C}}_t - \mathbf{C}_t$ and $\Delta_\Sigma := \hat{\boldsymbol{\Sigma}}_t - \boldsymbol{\Sigma}_t$. The hypothesis $B \geq 32\, M^4 \log(4K/\delta)/\lambda^2$ ensures $\varepsilon_B \leq \lambda/2$, so $\|\boldsymbol{\Sigma}_t^{-1}\Delta_\Sigma\| \leq 1/2$. The Neumann expansion of $(\boldsymbol{\Sigma}_t + \Delta_\Sigma)^{-1}$ yields

$$\hat{\boldsymbol{\Sigma}}_t^{-1} = \boldsymbol{\Sigma}_t^{-1} - \boldsymbol{\Sigma}_t^{-1}\Delta_\Sigma\boldsymbol{\Sigma}_t^{-1} + \mathbf{R}, \qquad \|\mathbf{R}\| \leq \frac{2\,\varepsilon_B^2}{\lambda^3}. \tag{84}$$

Substituting and isolating leading terms,

$$\hat{\mathbf{C}}_t\hat{\boldsymbol{\Sigma}}_t^{-1} - \mathbf{C}_t\boldsymbol{\Sigma}_t^{-1} = \Delta_C\,\boldsymbol{\Sigma}_t^{-1} - \mathbf{C}_t\,\boldsymbol{\Sigma}_t^{-1}\Delta_\Sigma\,\boldsymbol{\Sigma}_t^{-1} + \mathbf{E}, \tag{85}$$

where the residual $\mathbf{E}$ collects the second-order cross terms together with $(\mathbf{C}_t + \Delta_C)\,\mathbf{R}$. Using $\|\Delta_C\|, \|\Delta_\Sigma\| \leq \varepsilon_B$ and $\|\mathbf{C}_t\| \leq M^2$ gives $\|\mathbf{E}\| \leq 4\,M^2\varepsilon_B^2/\lambda^3$.

Taking trace and applying $|\operatorname{Tr}(A)| \leq K\,\|A\|_{\mathrm{op}}$ together with submultiplicativity of the operator norm,

$$\big|\hat{J}_t(f) - J_t(f)\big| \leq K\left(\frac{\|\Delta_C\|}{\lambda} + \frac{\|\mathbf{C}_t\|\,\|\Delta_\Sigma\|}{\lambda^2} + \frac{4\,M^2\varepsilon_B^2}{\lambda^3}\right). \tag{86}$$

Since $\mathbf{C}_t = \mathbb{E}[\mathbf{H}_i]$ with $\|\mathbf{H}_i\| \leq M^2$, we have $\|\mathbf{C}_t\| \leq M^2$. Substituting (83) into (86),

$$\big|\hat{J}_t(f) - J_t(f)\big| \leq \frac{K\,\varepsilon_B}{\lambda} + \frac{K\,M^2\,\varepsilon_B}{\lambda^2} + \frac{4\,K\,M^2\,\varepsilon_B^2}{\lambda^3} \leq \frac{2\,K\,M^2\,\varepsilon_B}{\lambda^2}\left(1 + \frac{2\,\varepsilon_B}{\lambda}\right), \tag{87}$$

where the second inequality uses $\lambda \leq M^2$ (which follows from $\boldsymbol{\Sigma}_t \preceq M^2\,\mathbf{I}$). Expanding $\varepsilon_B$ via (83) and absorbing the higher-order contributions into the second term yields the bound (81). $\qquad\square$

## A.6. Eigenfunctions for Tractable Priors

**Proposition A.13** (Eigenfunctions of $T_t T_t^*$ for Gaussian priors). *Let $X_0 \sim \mathcal{N}(\mathbf{0}, \boldsymbol{\Sigma})$ with eigendecomposition $\boldsymbol{\Sigma} = \sum_{k=1}^d \rho_k\,\mathbf{u}_k\mathbf{u}_k^\top$, $\rho_1 \geq \cdots \geq \rho_d > 0$, and let the forward process be $X_t = a_t X_0 + b_t\,\epsilon$ with $\epsilon \sim \mathcal{N}(\mathbf{0}, \mathbf{I})$ (in the DDPM parameterization, $a_t = \sqrt{\bar{\alpha}_t}$ and $b_t = \sqrt{1 - \bar{\alpha}_t}$). Then the linear functionals*

$$\phi_{t,k}(\mathbf{x}_t) := \mathbf{u}_k^\top\mathbf{x}_t, \qquad k = 1, \ldots, d, \tag{88}$$

*are eigenfunctions of the covariance operator $T_t T_t^*$, with eigenvalues*

$$\lambda_{t,k} = \frac{a_t^2\,\rho_k}{a_t^2\,\rho_k + b_t^2}. \tag{89}$$

*Proof.* Since $\phi_{t,k}$ is linear, both conditional expectations in the composition $T_t T_t^*$ reduce to matrix operations.

**Inner step ($T_t^*$).** Applying $T_t^*$ to $\phi_{t,k}$ and using linearity of conditional expectation,

$$(T_t^*\phi_{t,k})(\mathbf{x}_0) = \mathbb{E}\big[\mathbf{u}_k^\top X_t \mid X_0 = \mathbf{x}_0\big] = \mathbf{u}_k^\top\mathbb{E}[X_t \mid X_0 = \mathbf{x}_0] = a_t\,\mathbf{u}_k^\top\mathbf{x}_0. \tag{90}$$

**Outer step ($T_t$).** Applying $T_t$ to the result and using the standard Gaussian conditioning identity $\mathbb{E}[X_0 \mid X_t = \mathbf{x}_t] = a_t\,\boldsymbol{\Sigma}\,(a_t^2\,\boldsymbol{\Sigma} + b_t^2\,\mathbf{I})^{-1}\mathbf{x}_t$,

$$\begin{aligned}(T_t T_t^*\phi_{t,k})(\mathbf{x}_t) &= \mathbb{E}\big[a_t\,\mathbf{u}_k^\top X_0 \mid X_t = \mathbf{x}_t\big] \\ &= a_t^2\,\mathbf{u}_k^\top\boldsymbol{\Sigma}\,(a_t^2\,\boldsymbol{\Sigma} + b_t^2\,\mathbf{I})^{-1}\,\mathbf{x}_t. \end{aligned} \tag{91}$$

Since $\mathbf{u}_k$ is an eigenvector of both $\boldsymbol{\Sigma}$ (with eigenvalue $\rho_k$) and $(a_t^2\,\boldsymbol{\Sigma} + b_t^2\,\mathbf{I})^{-1}$ (with eigenvalue $(a_t^2\,\rho_k + b_t^2)^{-1}$), this simplifies to

$$(T_t T_t^* \phi_{t,k})(\mathbf{x}_t) = \frac{a_t^2\,\rho_k}{a_t^2\,\rho_k + b_t^2}\; \mathbf{u}_k^\top \mathbf{x}_t = \lambda_{t,k}\,\phi_{t,k}(\mathbf{x}_t). \tag{92}$$

$\square$

**Proposition A.14** (Eigenfunctions for the uniform prior on $S^1$). *Let $p_0$ be the uniform distribution on the unit circle $S^1 \subset \mathbb{R}^2$, parameterized as $\mathbf{x}_0(\theta) = (\cos\theta,\,\sin\theta)$ for $\theta \in [0, 2\pi)$, and let $p(\mathbf{x}_t \mid \mathbf{x}_0) = \mathcal{N}(\mathbf{x}_t;\,\sqrt{\bar\alpha_t}\,\mathbf{x}_0,\,(1 - \bar\alpha_t)\,\mathbf{I}_2)$. Then the eigenfunctions of $T_t^* T_t$ (equivalently, the right singular functions of $T_t$), expressed in angle coordinates, are the Fourier modes*

$$\psi_{t,1}(\theta) = 1, \qquad \psi_{t,2n}(\theta) = \sqrt{2}\,\cos(n\theta), \qquad \psi_{t,2n+1}(\theta) = \sqrt{2}\,\sin(n\theta), \qquad n = 1, 2, \ldots \tag{93}$$

*Proof.* We show that the kernel of $T_t^* T_t$ depends only on the angular difference, which identifies it as a convolution operator on $S^1$ and fixes the eigenfunctions as Fourier modes.

**Kernel of $T_t^* T_t$.** In angle coordinates, the operator $T_t^* T_t : L^2(\mu_0) \to L^2(\mu_0)$ is an integral operator with kernel

$$k(\theta, \theta') = \int_{\mathbb{R}^2} \frac{p(\mathbf{x}_t \mid \mathbf{x}_0(\theta))\,p(\mathbf{x}_t \mid \mathbf{x}_0(\theta'))}{p_t(\mathbf{x}_t)}\,d\mathbf{x}_t. \tag{94}$$

**Rotational invariance.** We claim that $k(\theta + \alpha,\,\theta' + \alpha) = k(\theta, \theta')$ for every $\alpha$, so that $k(\theta, \theta') = \kappa_t(\theta - \theta')$. Let $\mathbf{R}_\alpha \in \mathrm{SO}(2)$ denote rotation by angle $\alpha$ in $\mathbb{R}^2$. Since $\mathbf{x}_0(\theta + \alpha) = \mathbf{R}_\alpha\,\mathbf{x}_0(\theta)$ by definition of the angle parameterization, the change of variables $\mathbf{x}_t = \mathbf{R}_\alpha\,\mathbf{u}$ (with $d\mathbf{x}_t = d\mathbf{u}$, as rotations preserve Lebesgue measure) gives

$$k(\theta + \alpha,\,\theta' + \alpha) = \int_{\mathbb{R}^2} \frac{p(\mathbf{R}_\alpha\mathbf{u} \mid \mathbf{R}_\alpha\mathbf{x}_0(\theta))\,p(\mathbf{R}_\alpha\mathbf{u} \mid \mathbf{R}_\alpha\mathbf{x}_0(\theta'))}{p_t(\mathbf{R}_\alpha\mathbf{u})}\,d\mathbf{u}. \tag{95}$$

We verify each factor separately.

*Forward kernel.* Since $\mathbf{R}_\alpha$ is an isometry, $\|\mathbf{R}_\alpha\mathbf{u} - \sqrt{\bar\alpha_t}\,\mathbf{R}_\alpha\mathbf{v}\| = \|\mathbf{u} - \sqrt{\bar\alpha_t}\,\mathbf{v}\|$, and therefore

$$p(\mathbf{R}_\alpha\mathbf{u} \mid \mathbf{R}_\alpha\mathbf{x}_0(\theta)) = \mathcal{N}(\mathbf{R}_\alpha\mathbf{u};\,\sqrt{\bar\alpha_t}\,\mathbf{R}_\alpha\mathbf{x}_0(\theta),\,(1 - \bar\alpha_t)\,\mathbf{I}_2) = p(\mathbf{u} \mid \mathbf{x}_0(\theta)). \tag{96}$$

*Marginal.* Using the isometry property and the uniformity of $p_0 = 1/(2\pi)$,

$$p_t(\mathbf{R}_\alpha\mathbf{u}) = \frac{1}{2\pi} \int_0^{2\pi} p(\mathbf{R}_\alpha\mathbf{u} \mid \mathbf{x}_0(\theta''))\,d\theta'' = \frac{1}{2\pi} \int_0^{2\pi} p(\mathbf{R}_\alpha\mathbf{u} \mid \mathbf{R}_\alpha\,\mathbf{x}_0(\theta'' - \alpha))\,d\theta''$$

$$= \frac{1}{2\pi} \int_0^{2\pi} p(\mathbf{u} \mid \mathbf{x}_0(\theta'' - \alpha))\,d\theta'' = p_t(\mathbf{u}), \tag{97}$$

where the last step uses $2\pi$-periodicity of the integrand.

Substituting into (95),

$$k(\theta + \alpha,\,\theta' + \alpha) = \int_{\mathbb{R}^2} \frac{p(\mathbf{u} \mid \mathbf{x}_0(\theta))\,p(u \mid \mathbf{x}_0(\theta'))}{p_t(\mathbf{u})}\,d\mathbf{u} = k(\theta, \theta'). \tag{98}$$

Hence $k(\theta, \theta') = \kappa_t(\theta - \theta')$, with $\kappa_t$ defined by

$$\kappa_t(\Delta\theta) := \int_{\mathbb{R}^2} \frac{p(\mathbf{x}_t \mid \mathbf{x}_0(0))\,p(\mathbf{x}_t \mid \mathbf{x}_0(\Delta\theta))}{p_t(\mathbf{x}_t)}\,d\mathbf{x}_t. \tag{99}$$

**Convolution structure and eigenfunctions.** Since $k(\theta, \theta') = \kappa_t(\theta - \theta')$, the operator $T_t^* T_t$ acts on $L^2(S^1)$ as circular convolution:

$$(T_t^* T_t f)(\theta) = \int_0^{2\pi} \kappa_t(\theta - \theta') f(\theta') \frac{d\theta'}{2\pi}. \tag{100}$$

By the spectral theorem for convolution operators on $S^1$, the eigenfunctions are the Fourier modes $\{1, \cos(n\theta), \sin(n\theta)\}_{n \geq 1}$. Each $\cos(n\theta)$ and $\sin(n\theta)$ share the same eigenvalue $\lambda_{t,n}$ because $\kappa_t$ is real and even (the latter following from $k(\theta, \theta') = k(\theta', \theta)$ by symmetry of (94)). The $\sqrt{2}$ normalization in (93) ensures $\|\psi_{t,k}\|_{\mu_0} = 1$. $\qquad\square$

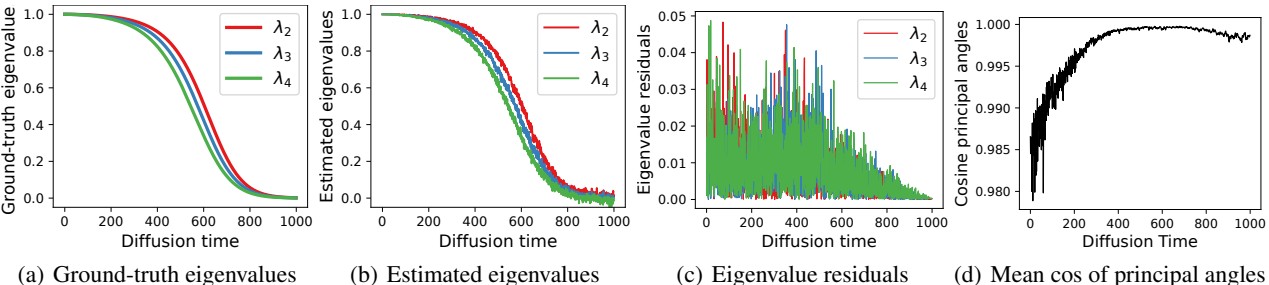

(a) Ground-truth eigenvalues     (b) Estimated eigenvalues     (c) Eigenvalue residuals     (d) Mean cos of principal angles

*Figure 8.* **Spectral recovery on a centered Gaussian prior in $\mathbb{R}^{20}$.** (a) Closed-form ground-truth eigenvalues of $T_t T_t^*$ from Proposition A.13. (b) Monte-Carlo estimates produced by $f_\phi$. (c) Absolute residuals between (a) and (b). (d) Mean cosine of the principal angles between the true and estimated leading $K = 3$ eigenspaces.

## B. Illustrative Examples

### B.1. Gaussian Prior

We demonstrate our spectral learning framework on a centered Gaussian prior, where Proposition A.13 gives closed-form expressions for both the eigenvalues and eigenfunctions of $T_t T_t^*$, allowing direct comparison with our estimates.

**Setup.** The prior $p_0$ is a centered Gaussian on $\mathbb{R}^{20}$ whose covariance spectrum has a geometric decay $\{40 \cdot 0.7^{k-1}\}_{k=1}^{20}$. The forward process follows a linear DDPM schedule with $T = 1000$ timesteps. We train the eigenfunction network $f_\phi$ (architecture and loss as described in §4.3, with ridge $\xi = 0.001$) to recover the leading $K = 3$ non-trivial eigenfunctions of $T_t T_t^*$, excluding the constant mode (the eigenfunction $\phi_{t,1} \equiv 1$ with eigenvalue 1).

**Results.** Fig. 8 reports four diagnostics over diffusion time. Fig. 8(a) shows the leading three ground-truth eigenvalues $\lambda_{t,2}, \lambda_{t,3}, \lambda_{t,4}$ from Eq. (89), and Fig. 8(b) the corresponding Monte-Carlo estimates produced by $f_\phi$; the two curves match closely along the entire trajectory. Fig. 8(c) plots the absolute residuals, which remain below 0.06 and decay as the eigenvalues approach zero at large $t$. Fig. 8(d) shows the mean cosine of the principal angles between the true leading eigenspace and our estimate: it rises from 0.97 at small $t$ to essentially 1 at large $t$. This is consistent with the structure of the operator: at small $t$, $T_t T_t^*$ is close to identity and the eigengap is small, so the leading eigenspace is weakly identified, while over the second half of the trajectory the cosine stays above 0.995, indicating near-perfect subspace recovery.

### B.2. Eigenfunction Visualization

To build intuition for the singular functions learned by the spectral framework, we visualize the right singular functions of $T_t$ on four simple priors: the unit circle $\mathcal{S}^1 \subset \mathbb{R}^2$, the uniform square $[-0.5, 0.5]^2$, the unit disk $\{x \in \mathbb{R}^2 : \|x\| \leq 1\}$, and an annulus $\{x \in \mathbb{R}^2 : r_{\text{in}} \leq \|x\| \leq r_{\text{out}}\}$. In all cases, we train the eigenfunction network as described in §4.3 and visualize the learned right singular functions $\psi_{t,k}(x_0)$ by coloring each sample $x_0$ from the prior according to the function value $\psi_{t,k}(x_0)$.

For the **circle prior**, the support is a one-dimensional manifold, and the right singular functions of $T_t$ are the Fourier basis (Proposition A.14). The learned modes, depicted in Figure 9(a), recover this structure: $\psi_{t,2}$ exhibits two nodal points, and subsequent pairs display progressively higher-frequency oscillations along the circle (the constant eigenfunction is not shown). Modes appear in degenerate pairs, consistent with the rotational symmetry of the prior.

For the **square prior**, the learned modes, shown in Figure 9(b), recover axis-aligned oscillatory patterns of increasing spatial frequency, akin to the Fourier basis.

For the **disk prior**, the support is a two-dimensional manifold with a circular boundary. The learned modes, shown in Figure 9(c), separate into radial and angular components: angular oscillations along $\theta$ are modulated by a radial envelope. The rotational symmetry of the prior again yields degenerate pairs of angular modes.

For the **annulus prior**, the learned modes, shown in Figure 9(d), again factor into radial and angular components, but the inner boundary modifies the radial envelope relative to the disk and the angular modes wrap continuously around the hole. Rotational symmetry continues to enforce degenerate pairs.

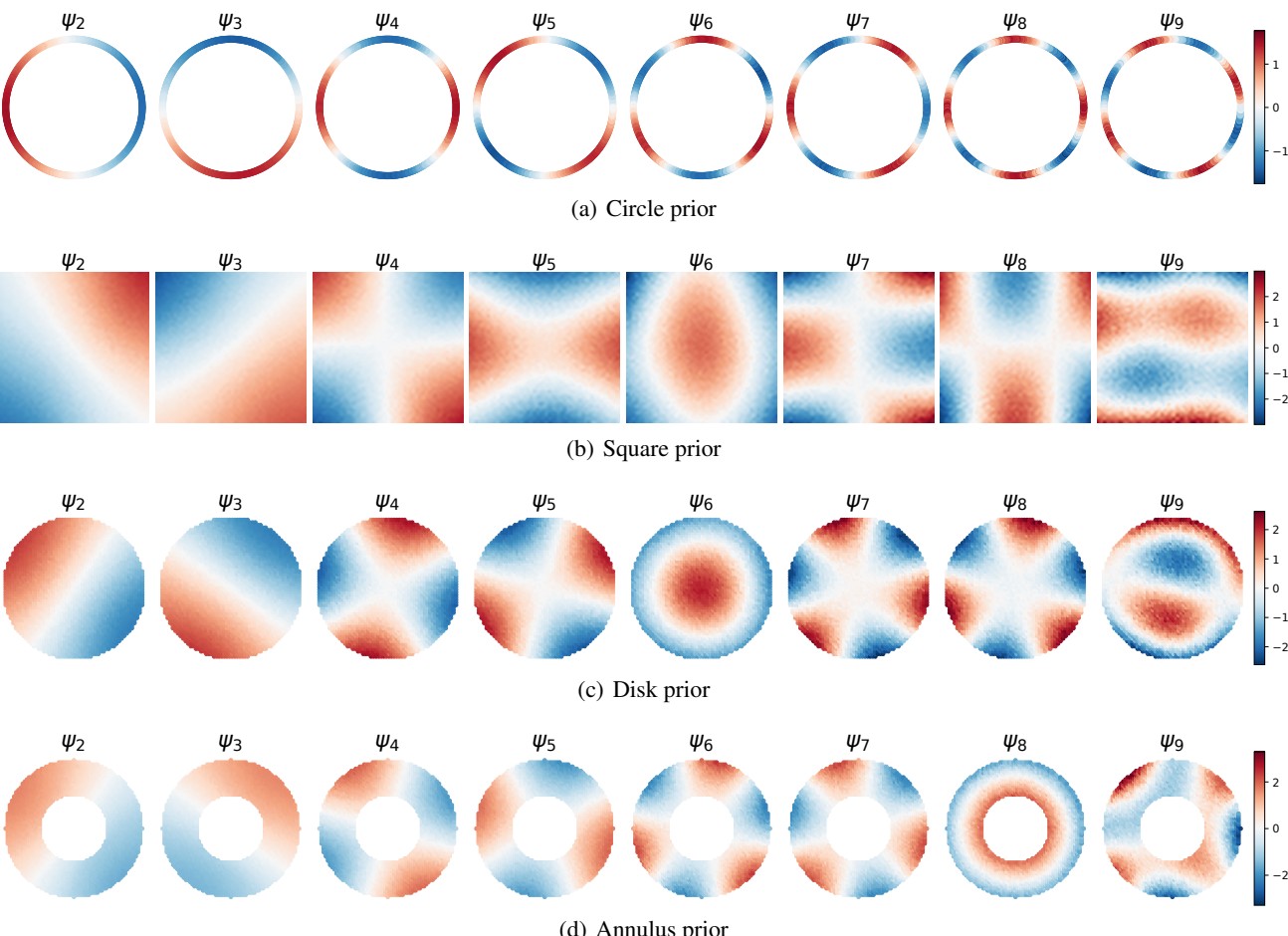

*Figure 9.* **Learned singular functions on simple manifolds** ($t = 100$). Each row shows the leading 8 right singular functions $\psi_{t,k}$, with samples from the prior colored by function value. On the unit circle, the modes recover the Fourier basis on $\mathcal{S}^1$ (Proposition A.14). On the uniform square, they recover axis-aligned oscillations of increasing spatial frequency. On the disk, modes factor into a radial envelope and angular harmonics. On the annulus, the additional inner boundary modifies the radial envelope and the angular modes wrap around the hole. In all cases, spatially smooth modes appear first, and rotationally symmetric priors yield degenerate angular pairs.

In all four cases, low-frequency (spatially smooth) modes appear first, corresponding to large singular values, while high-frequency modes appear later with diminishing singular values.

# C. Experiments

## C.1. Training Setup

We employed a consistent backbone architecture across all datasets. The networks $f_\phi$ are parameterized as ResNets, incorporating time modulation via FiLM blocks to condition on the diffusion timestep.

**Datasets and base models.**

- **CIFAR-10.** We trained $f_\phi$ with an output dimension of $K = 512$ on the standard train split, comprising 50,000 $32 \times 32$ images across 10 classes. For the diffusion backbone, we used the pre-trained unconditional DDPM pipeline `google/ddpm-cifar10-32`, available on HuggingFace.

- **CelebA-HQ.** We trained $f_\phi$ with an output dimension of $K = 512$ on the full dataset of 30,000 images, resized to $256 \times 256$. We used the pre-trained unconditional DDPM pipeline `google/ddpm-ema-celebahq-256`, available on HuggingFace.

- **ImageNet.** We trained $f_\phi$ with an output dimension of $K = 2000$ on the training split comprising 1,281,167 images, resized to $64 \times 64$. We used a $64 \times 64$ improved-diffusion unconditional DDPM pipeline available on GitHub[1].

**Optimization.** Training was performed using the Adam optimizer, with the loss described in Algorithm 1. We used an initial learning rate of $10^{-4}$, which was decayed exponentially by a factor of 0.995 every epoch. The ridge parameter to compute the batch whitening matrices $\mathbf{W}_t$ was set to $\xi = 0.001$ in all experiments. Training on CIFAR-10 and CelebA-HQ was conducted on a single NVIDIA GPU with 48GB of memory (batch size of 2048) while ImageNet training was performed on 4 NVIDIA GPUs with 48GB memory (batch size per GPU of 4096).

## C.2. Label Guidance

**CIFAR-10.** We generated 2,650 samples per class with a DDIM sampler (100 timesteps, $\eta = 1.0$). We used a guidance strength of $\kappa = 10$ on Spectral Guidance and the implementations and hyperparameters of all baselines from Ye et al. (2024). Guidance was evaluated across all 10 classes, using an external ConvNext classifier[2]. We report the average top-1 accuracy.

**CelebA-HQ.** Following the benchmarks established by Ye et al. (2024), we evaluated compositional attribute guidance. We tested two categories of combinations:

1. **Gender $\times$ Hair Color:** (Male/Female) $\times$ (Black/Blond Hair)

2. **Gender $\times$ Age:** (Male/Female) $\times$ (Young/Old)

We generated 256 samples per combined target with a DDIM sampler (100 timesteps, $\eta = 1.0$). We used a guidance strength of $\kappa = 10$ on Spectral Guidance. For the baselines, we used the the implementations[3] and hyperparameters from Ye et al. (2024). Accuracy was computed using three external classifiers for Gender[4], Age[5], and Hair Color[6]. A generated sample is considered correct only if it satisfies both target attributes simultaneously.

**ImageNet.** We evaluated guidance of 4 labels: kuvasz, hamster, bicycle-built-for-two and nematode, following Ye et al. (2024). We generated 256 images per label with a DDIM sampler (400 timesteps, $\eta = 1.0$). We used a guidance strength of $\kappa = 30$ on Spectral Guidance. Guidance validity was evaluated with a pre-trained DeiT from HuggingFace[7]. We report the average top-1 accuracy.

---

[1]https://github.com/openai/improved-diffusion
[2]https://huggingface.co/ahsanjavid/convnext-tiny-finetuned-cifar10
[3]https://github.com/YWolfeee/Training-Free-Guidance
[4]https://huggingface.co/rizvandwiki/gender-classification
[5]https://huggingface.co/ibombonato/swin-age-classifier
[6]https://huggingface.co/londe33/hair_v02
[7]https://huggingface.co/facebook/deit-small-patch16-224

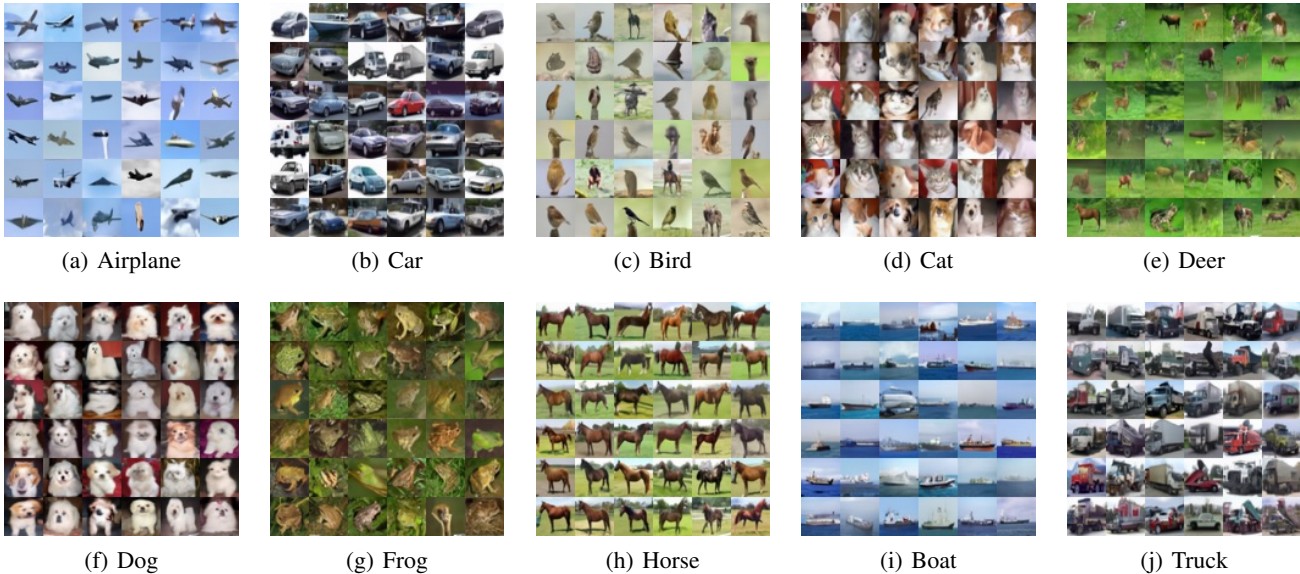

| (a) Airplane | (b) Car | (c) Bird | (d) Cat | (e) Deer |

| (f) Dog | (g) Frog | (h) Horse | (i) Boat | (j) Truck |

*Figure 10.* CIFAR-10 label-conditioned samples generated via Spectral Guidance ($K = 512$, $\xi = 0.001$, $\kappa = 10.0$).

### C.3. CLIP Guidance

We benchmarked open-vocabulary guidance on CelebA-HQ using 15 natural language prompts, detailed in Table 4. These prompts encompass attributes from the original dataset as well as new features such as pose and background. We used the TFG (Ye et al., 2024) implementation of all baselines with the following hyperparameters, found via grid search:

- **DPS.** Guidance strength $= 10.0$

- **LGD.** Guidance strength $= 400.0$; Number of samples to estimate posterior: 10

- **FreeDoM.** Guidance strength $= 20.0$; $N_{\text{recur}} = 2$

- **MPGD.** Guidance strength $= 20.0$;

- **UGD.** Guidance strength $= 20.0$; $N_{\text{iter}} = 5$; $N_{\text{recur}} = 1$

- **TFG.** $\rho = 30.0$; $\mu = 1.0$, $N_{\text{recur}} = 2$, $N_{\text{iter}} = 5$, $\bar{\gamma} = 0.001$, Number of samples to estimate posterior: 1

We used a guidance strength of $\kappa = 20.0$ for Spectral Guidance. For all methods, guidance was computed using CLIP ViT-B/32, via the cosine similarity between the predicted clean image and the target text prompt. We generated 100 images per prompt with a DDIM sampler (100 timesteps, $\eta = 1.0$) and report the average VQAScore using `llava-v1.5-7b`.

### C.4. Mask Guidance

In the mask guidance experiment, we used 1,000 binary hair masks ($256 \times 256$) from CelebAMask-HQ to guide the generation. Spectral Guidance uses the same model $f_\phi$ and precomputed singular functions $\{\Phi_t\}_{t \in \mathcal{T}}$ from the CLIP and label guidance experiments on CelebA-HQ. In contrast, all baselines rely on the BiSeNet[8] segmentation model for guidance. We used the TFG (Ye et al., 2024) implementation of all baselines with the following hyperparameters:

- **DPS.** Guidance strength $= 3.0$

- **LGD.** Guidance strength $= 400.0$; Number of samples to estimate posterior: 5

- **FreeDoM.** Guidance strength $= 30.0$; $N_{\text{recur}} = 1$

---

[8] https://github.com/zllrunning/face-parsing.pytorch

*Table 4.* **Prompts used for CLIP Guidance.**

| Prompt |
| --- |
| a man wearing sunglasses |
| a dark haired man with a beard |
| man with a goatee |
| old man wearing sunglasses |
| a woman with dark hair wearing sunglasses |
| a woman with red hair |
| a person with messy hair |
| person with their mouth open |
| person against a white background |
| a young woman wearing a hat |
| a studio headshot |
| a woman wearing eye shadow |
| three-quarter profile portrait |
| a woman with brown hair smiling |
| a man with a receding hairline and a beard |

- **MPGD.** Guidance strength $= 30.0$;

- **UGD.** Guidance strength $= 50.0$; $N_{\text{iter}} = 5$; $N_{\text{recur}} = 1$

- **TFG.** $\rho = 30.0$; $\mu = 1.0$, $N_{\text{recur}} = 1$, $N_{\text{iter}} = 5$, $\bar{\gamma} = 0.001$, Number of samples to estimate posterior: 1

We generated one image per target mask with a DDIM sampler (100 timesteps, $\eta = 1.0$). Guidance validity was evaluated by feeding the generated image to an independent face parsing model from HuggingFace[9] and computing the IoU between the detected hair mask and the target one. Fidelity was assessed via the $\log$ KID between the generated images and 1,000 random images from the dataset.

---

[9]https://huggingface.co/jonathandinu/face-parsing

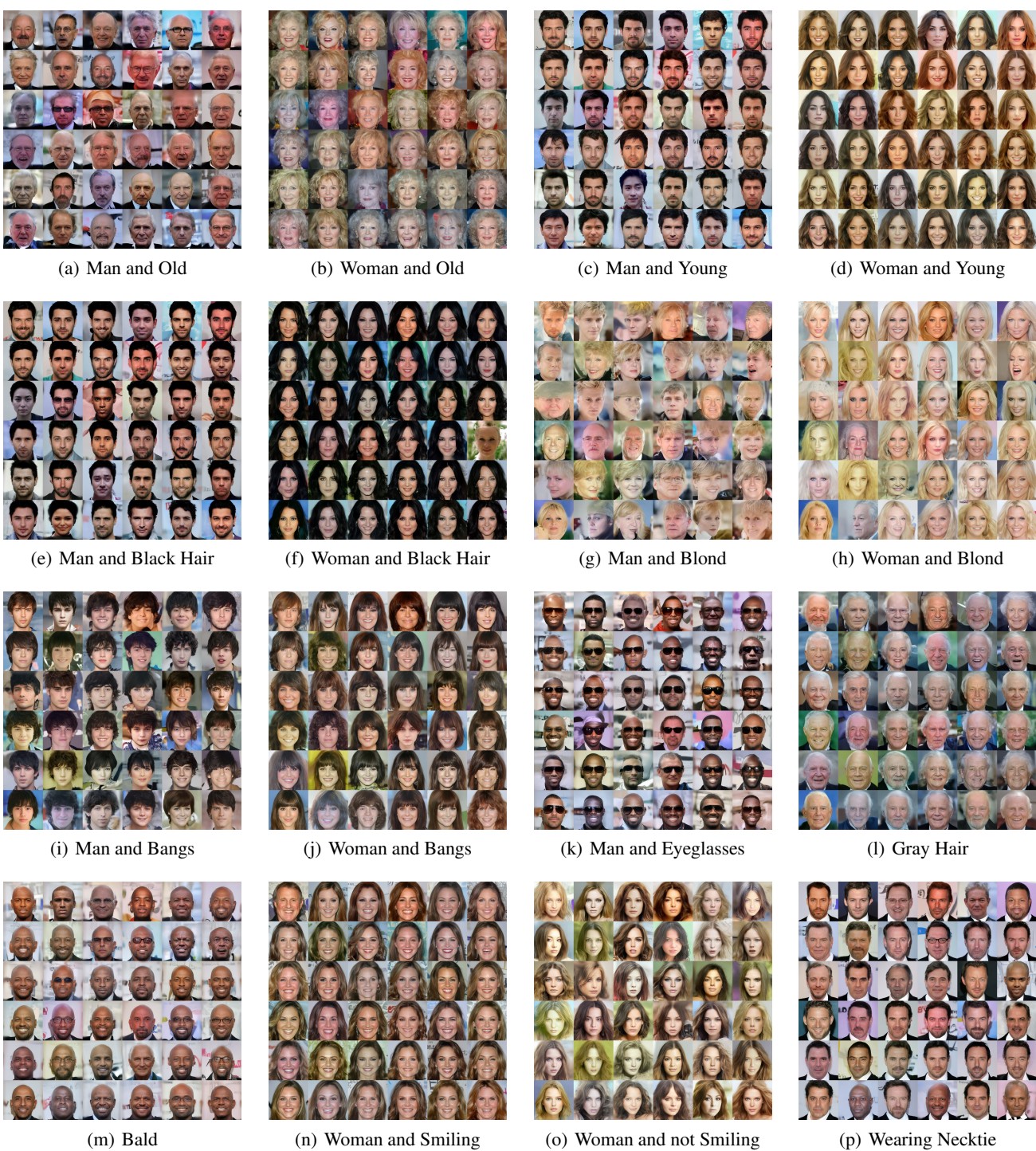

(a) Man and Old  (b) Woman and Old  (c) Man and Young  (d) Woman and Young

(e) Man and Black Hair  (f) Woman and Black Hair  (g) Man and Blond  (h) Woman and Blond

(i) Man and Bangs  (j) Woman and Bangs  (k) Man and Eyeglasses  (l) Gray Hair

(m) Bald  (n) Woman and Smiling  (o) Woman and not Smiling  (p) Wearing Necktie

*Figure 11.* CelebA-HQ attribute conditioned samples generated with Spectral Guidance ($K = 512, \xi = 0.001, \kappa = 10.0$).

