# OpenReview forum: "Spectral Guidance for Flexible and Efficient Control of Diffusion Models"
_ICML.cc/2026/Conference — ICML 2026 regular_

### Official Review · Reviewer_ptT8 · 2026-02-26

**Soundness:** 3
**Presentation:** 3
**Significance:** 2
**Originality:** 3
**Overall Recommendation:** 4
**Confidence:** 3

**Summary:**

The paper propose a spectral decomposition guidance method for diffusion models. The authors define an operator from clean space $0$ to a noisy state $t$ and another operator modeling the reverse. By studying the spectrum of these operators, the authors identify a set of associated singular functions as being central to diffusion guidance. Empirically, the authors propose to learn a subset of these functions via deep learning and a self-supervised objective, claiming that this is sufficient in practice. Results are reported for label-guidance (CIFAR-10), attribute-guidance and CLIP guidance (CelebA-HQ). Some analysis is also given regarding the optimal guidance window and phase transitions in diffusion.

**Compliance With Llm Reviewing Policy:**

Affirmed.

**Final Justification:**

The proposed method is interesting both theoretically and practically. The rebuttal added larger scale experiments, clarified limitations of the method in inverse problems and studied analytically tractable scenarios. I am increasing my score conditional on the authors being explicit about the limitations of their method in the final version of the paper, including the additional experiments ran during rebuttal and updating the paper with the analysis in the tractable case.

**Key Questions For Authors:**

1. Regarding W1: can the authors show results for inverse problems? For example, how does the proposed method compare to standard super-resolution and deblurring tasks?

2. Regarding W2: can the authors demonstrate results for larger scale and more diverse settings, e.g., ImageNet? Judging from Algorithm 1, the resource requirements do not appear to change with the diffusion model size so I deem this a reasonable ask.

3. It would be helpful if the authors could include some discussion around stability of the self-supervised objective. Do you observe collapse? It would also be useful (but perhaps not as simple) to gain a more fundamental understanding of the singular functions. For example, given analytically tractable distributions, like Gaussians, does your training result in the correct functions?

**Limitations:**

The paper does not explicitly discuss limitations. I encourage the authors to include a Limitations section in the final version of the paper. For example, the authors have demonstrated their method on relatively simple settings (see W2).

**Strengths And Weaknesses:**

Strengths
-

1. The paper easy to read and generally well-written.
2. To my knowledge, the theoretical analysis offers a new lens on guidance methods.
3. The results in Table 2 and Figure 9 largely validate the method.

Weaknesses
-

1. The paper cites and compares DPS on various tasks. However, DPS is originally an inverse problem approach. The paper does not give results for standard inverse problem benchmarks, which are well-established in the literature.

2. The method is benchmarked on fairly simple and low-resolution datasets. The paper could be strengthened with results from more modern and higher-resolution models.

3. Stability of the self-supervised objective is not discussed (e.g., collapse to all zeros) and it is unclear whether the method is applicable to larger scale settings.

---

> ### Author Rebuttal · Authors · 2026-03-31
>
> We thank the reviewer for the valuable feedback and suggestions.
>
> **Inverse Problems (W1, Q1).** We did not conduct inverse problem benchmarks, as our low-rank approximation is theoretically designed for semantic guidance. Inverse problems require approximating a posterior that is concentrated on a small region of the data manifold, requiring a very high-rank approximation. Our approach deliberately filters out high-frequency modes to isolate the stable, semantic structure of the diffusion operator. We view thus semantic and inverse-problem guidance as complementary regimes, and note that posterior-mean methods are well-suited to linear inverse problems precisely because the affine structure makes the point estimate exact, unlike semantic tasks, where the loss is highly nonlinear. We will add this to the paper.
>
> **Additional Results on More Diverse Settings (W2, Q2).** Following the reviewer’s comment that more diverse and complex datasets would strengthen the paper, we have conducted new experiments on ImageNet guidance. We followed the experimental setup from CIFAR-10 and CelebA-HQ, training our $f_\phi$ network with $K=2000$, on ImageNet, following Algorithm 1 (ridge $\xi=10^{-3}$). We report the guidance results for the 4 labels of the TFG paper:
>
> **Table: ImageNet Guidance Results**
> | Method | DPS  | LGD  | FreeDoM | MPGD | UGD  | TFG  | Ours |
> | ---------------- | ---- | ---- | ------- | ---- | ---- | ---- | ---- |
> | **Accuracy (%)** | 38.8 | 11.5 | 19.7    | 6.80 | 25.5 | 40.9 | **41.6** |
>  | **FID ↓**  | 193  | 210  | 200     | 239  | 205  | 176  | **183**  |
>
> Two points are worth noting. First, TFG is a framework that subsumes all others under a unified hyperparameter search. Matching TFG therefore means matching the best achievable combination of all training-free methods. Second, the spectral transition on ImageNet (like that observed for CIFAR-10 and CelebA-HQ in Fig. 6) occurs in the final 15% timesteps of the forward diffusion trajectory. Within this short window, matching the best combined baseline confirms that the representational quality of $f_\phi$​ scales to complex datasets.
>
> **Stability and Collapse (W3).** Training stability and collapse prevention relies on the whitening being well-conditioned. We conducted an analysis of the evolution of the minimum eigenvalue of the covariance $\lambda_{\min}(C_{\Phi\Phi})$. If $\lambda_{\min}(C_{\Phi\Phi}) \gg \xi$ (ridge), all $K$ dimensions carry signal and no collapse occurs.
>
> **Table: Evolution of  the ratio $\lambda_{\min}(C_{\Phi\Phi}) / \xi$ during training**
> | Dataset | Epoch 1 | Epoch 10 | Epoch 50 | Epoch 100 |
> | ------------------------------- | ------- | -------- | -------- | --------- |
>  | **ImageNet ($\xi = 10^{-3}$)**  | 0.01    | 120      | 8700     | 64000     |
> | **CIFAR-10 ($\xi = 10^{-6}$)**  | 0.0002  | 0.2      | 30       | 180       |
>
> Across all datasets, at initialization, $\lambda_{\min}(C_{\Phi\Phi})$ is low. As training progresses, the network populates all $K$ dimensions, with the ratio stabilizing above 1.
>
> **Understanding Singular Functions for Tractable Distributions (Q3).** This is an excellent suggestion. We will add a new section showing exact recovery for synthetic Gaussian priors. For $X_0\\sim \\mathcal{N}(0,\\Sigma)$, with eigenpairs of $\\Sigma$ being $(\\lambda_k, u_k)$, ​the linear functionals $\phi_{t,k}(x_t)=u_k^\top x_t$ are eigenfunctions of $T_t T_t^\ast$, with eigenvalue
>
> $$
> \frac{\bar{\alpha}_t \lambda_k}{\bar{\alpha}_t \lambda_k + (1-\bar{\alpha}_t)}
> $$
>
> In this setting, our loss (Eq. 23) reduces to the generalized Rayleigh quotient, whose solution is exactly $u_k$. Thus, our training provably recovers the correct eigenfunctions. We validated this empirically on a synthetic 20-d Gaussian prior, comparing the learned spectrum with the ground-truth via Mean Absolute Error (MAE) and eigenfunctions via the mean cosine alignment:
>
> **Table: Spectral Recovery for Gaussian Priors (t=diffusion time for $t=1,\dots,1000$)**
> | Metric  | t = 1  | t = 50 | t = 100 | t = 500 | t = 750 | t = 1000 |
> |---------- | ------ | ------ | ------- | ------- | ------- | -------- |
> | **Eigenvalue MAE ↓**                          | 0.0005 | 0.002  | 0.01    | 0.02    | 0.02    | 0.02     |
> | **Subspace Alignment (mean cos) ↑** | 1.0    | 1.0    | 1.0     | 1.0     | 1.0     | 1.0      |
>
> **Limitations.** Following the reviewer's suggestion, we have added a Limitations section to the paper. Our method has been evaluated on pixel-space diffusion models. Extending to latent diffusion models such as Stable Diffusion is theoretically straightforward but left to future work. The offline training phase, while amortized over future generations, introduces an upfront cost when applying the method to a new data distribution. Finally, as discussed in the inverse problems response, the low-rank projection is designed for semantic guidance and is does target pixel-level reconstruction tasks.

---

> > ### Author Rebuttal · Reviewer_ptT8 · 2026-04-03
> >
> > Thank you for your responses.
> >
> > I remain of the opinion that testing on inverse problems is valuable as it does fall within the scope of guidance and flexibility is something that is advertised in the paper. In fact, you already have qualitative mask guidance results (Fig. 4), which is essentially an inpainting inverse problem. Given the annotated masks, it is unclear to me why DPS would require access to a segmentation model in that case (line 373).
> >
> > Perhaps your method does not perform as well there but it is important to identify limitations and trade-offs. Do you expect that DPS might more accurately track the measurements at the cost of worse perceptual scores (e.g., FID) whereas your method might do the opposite?
> >
> > Moreover, though you imply that comparison with DPS would not be fair as it was designed for a different problem, you still use it in your qualitative analysis on semantic tasks as a baseline (e.g., Fig. 3).

---

> > > ### Author Response · Authors · 2026-04-07
> > >
> > > We thank the reviewer for the follow-up remarks. We address the core points regarding the mask experiment, the inverse problem trade-off, and our baselines.
> > >
> > > **Clarification of the Mask Guidance Experiment.** We believe there may be a misunderstanding about the mask guidance experiment in our paper, which we hope to clarify. Our experiment is *not* an inverse problem like inpainting. The task is: given a target hair boundary mask $M\_\\text{target}$, generate a face whose hair region follows that shape. There are no observed pixels to condition on. Instead, the mask specifies the desired semantic layout for the image, not a partial observation of a ground-truth image.
> > >
> > > This distinction explains why DPS requires a segmentation model here. To guide generation, DPS requires a differentiable measurement operator $A$ to compute the gradient of a loss $ l(M\_{\\text{target}}, A(\\hat{x}_0))$, where $\\hat{x}_0$ is the posterior mean estimate. Because $M\_{\\text{target}}$ is a mask, $A$ must be a function that takes an image and outputs a hair mask, hence the need for a hair segmentation model. Without this auxiliary network, DPS has no differentiable way to enforce the mask layout. In contrast, our approach approximates the conditional expectation $\\mathbb{E}[M \\mid x\_t]$ without any auxiliary model.
> > >
> > > **Inverse problems.** In a linear inverse problem $y=Ax\_0+\\eta$, the goal is to sample from the posterior $p(x\_0\\mid y)$, which requires $Ax\_0\\approx y$ to be satisfied at the pixel level. Our method approximates the conditional operator by projecting it onto the span of its top-$K$ eigenfunctions. By restricting the generation trajectory to this low-dimensional subspace, we inherently lack the degrees of freedom required to satisfy a large number of independent pixel-wise constraints, preventing exact enforcement.
> > >
> > > Consider an inpainting task on a $256\\times 256\\times 3$ image (e.g. CelebA-HQ) where half the pixels are constrained. This imposes $C=98,304$ independent linear constraints. Our method restricts generation to the span of the top $K=512$ eigenfunctions. Because $K\\ll C$, the system is massively overdetermined, making it unsuitable to drive the error $\\|y-A\\hat{x}\_0\\|_2\to 0$.
> > >
> > > Therefore, as the reviewer hypothesized, we expect DPS to satisfy hard pixel-level measurements (like exact boundary pixels in inpainting) at the cost of perceptual quality, whereas our method does the opposite: producing semantically coherent structural layouts without exact pixel-level fidelity. We will explicitly mention this trade-off in the Limitations section.
> > >
> > > **Using DPS as a baseline.** We benchmark against DPS on semantic tasks not to claim it is optimal for them, but because its flexibility has made it the *de facto* standard baseline for training-free guidance, even for non-linear tasks. While DPS is theoretically principled for linear inverse problems, semantic tasks do not involve partial observations of a ground-truth image. Instead, they involve highly non-linear objectives. In these regimes, the gradient approximations that make DPS effective for linear tasks become ill-suited. In addition, applying DPS to semantic guidance requires a pre-trained auxiliary model (such as a segmentation model, CLIP, or a classifier) to formulate the differentiable measurement operator.
> > >
> > > Comparing against DPS motivates our low-rank approach as an alternative tailored for the domain of semantic guidance, where exact affine measurement operators are unavailable and reliance on auxiliary networks incurs sampling overhead.

---

### Official Review · Reviewer_QQaf · 2026-03-13

**Soundness:** 3
**Presentation:** 3
**Significance:** 3
**Originality:** 3
**Overall Recommendation:** 4
**Confidence:** 3

**Summary:**

This paper introduces a framework for controlling the diffusion model for guided generation. It learns a frame of features that are robust to noise and then uses them for guidance. Empirical results show improved performance over training-free guidance methods.

**Compliance With Llm Reviewing Policy:**

Affirmed.

**Final Justification:**

After the author's rebuttal on ImageNet experiments and follow-up response, I find the method somewhat limited and only marginally better than the weaker version of the baseline method (see below). I maintain my original score of 4.

The authors report that their method on ImageNet achieves 41.6% accuracy and FID score of 183, which is marginally better than TFG's method 40.9% accuracy and FID score of 176. However, after checking the TFG paper, it appears that a stronger baseline, which achieves far better performance, is when setting the TFG recursion parameter to 4, which achieves 59.8% accuracy and an FID score of 165.

**Key Questions For Authors:**

What is the performance of the model on ImageNet or the fine-grained task in the TFG paper?

**Strengths And Weaknesses:**

Strength: The paper proposes a novel method for guiding diffusion model sampling. The inference process is more efficient than the training-free methods.
Weakness: The proposed method shows promising performance on CIFAR-10 and CelebA, but has not been tested on the ImageNet task with TFG, which has more object classes and more fine-grained features may be needed. The offline training of the model takes 10 hours on Celeba, which is quite long.

---

> ### Author Rebuttal · Authors · 2026-03-31
>
> We thank the reviewer for the constructive feedback.
>
> **ImageNet Experiments (W1,Q1).** We have conducted an additional experiment on ImageNet, following the experimental setup used in CIFAR-10 and CelebA-HQ. We trained our $f_\phi$ network with $K=2000$ (ridge $\xi=10^{-3}$), following Algorithm 1. We report the guidance benchmark results for the 4 ImageNet labels considered in the TFG paper:
>
> **Table: ImageNet Guidance Results**
> | Method           | DPS  | LGD  | FreeDoM | MPGD | UGD  | TFG  | Ours |
> | ---------------- | ---- | ---- | ------- | ---- | ---- | ---- | ---- |
> | **Accuracy (%)** | 38.8 | 11.5 | 19.7    | 6.80 | 25.5 | 40.9 | **41.6**    |
>  | **FID ↓**             | 193  | 210  | 200     | 239  | 205  | 176  | **183**    |
>
> Two points are worth noting. First, TFG is a framework that subsumes all others under a unified hyperparameter search. Matching TFG therefore means matching the best achievable combination of all training-free methods. Second, the spectral transition on ImageNet (like that observed for CIFAR-10 and CelebA-HQ in Fig. 6) occurs in the final 15% timesteps of the forward diffusion trajectory. Within this short window, matching the best combined baseline confirms that the representational quality of $f_\phi$​ scales to complex datasets.
>
> **Offline Training (W2).** The reviewer correctly identifies the one-time training cost as a trade-off. We argue it is a favorable one for three reasons:
> - Training-free methods backpropagate through the full U-Net at every sampling step. Generating a large dataset with these methods requires thus more total compute than our one-time training.
> - The training is unsupervised and task-agnostic: the same $f_\\phi$​, trained once, supports classifier guidance, CLIP guidance, mask guidance, or any other semantic task without retraining.
> - At inference our method requires only a forward pass through the lightweight $f_\\phi$​, yielding a per-step latency comparable to unconditional sampling and nearly 4x faster than TFG, on CelebA-HQ, as shown in Table 3.

---

> > ### Author Rebuttal · Reviewer_QQaf · 2026-04-03
> >
> > Thank you for your response and for adding the ImageNet experiment. Since the accuracy gain over the TFG baseline is marginal, the method's potential may be limited. I maintain my score.

---

> > > ### Author Response · Authors · 2026-04-07
> > >
> > > We thank the reviewer for their continued engagement, positive assessment, and for maintaining their score. We will explicitly include these ImageNet experiments in the final version to demonstrate that our approach scales to significantly more complex settings.
> > >
> > > Regarding the performance margin over the TFG baseline, we would like to highlight the context of these results:
> > >
> > > - Transitioning from datasets of 30k–60k images to ImageNet (>1 million images) within the short rebuttal time frame precluded any hyperparameter search.
> > > - TFG establishes a competitive upper bound by relying on a dataset-specific hyperparameter search to combine the strengths of prior methods (DPS, LGD, FreeDoM, MPGD, and UGD). In contrast, our reported ImageNet results were achieved using the same training and guidance settings from our CIFAR-10 and CelebA-HQ experiments.
> > >
> > > The fact that our method matches and slightly outperforms the tuned TFG baseline on their provided benchmark, validates the scalability of our approach to much more complex distributions.

---

### Official Review · Reviewer_Dri7 · 2026-03-13

**Soundness:** 2
**Presentation:** 2
**Significance:** 2
**Originality:** 2
**Overall Recommendation:** 4
**Confidence:** 3

**Summary:**

The authors propose Spectral Guidance as a technique for flexible control of conditional generation with an unconditional diffusion model. This is done by characterizing the singular functions of the conditional expectation operator and leveraging self supervised learning to learn them. This is motivated by the observation that only a low dimensional subspace of features is necessary for effective guidance during sampling. Experiments on CIFAR-10 and CelebA-HQ are performed for label guidance, attribute guidance, CLIP guidance, and mask guided generation. The authors also identify a phase transition in the generative process that hints at the most effective window for guidance.

**Compliance With Llm Reviewing Policy:**

Affirmed.

**Final Justification:**

My recommendation is **Weak Accept**, contingent on the authors incorporating all promised revisions into the final manuscript.

**Resolved concerns:**

The compactness proof is now correct and complete. The vanishing singular values result is now correctly restricted to zero-mean $f \in L^2(p_0)$. The Eckart-Young connection is acknowledged for the truncation bound.

**Partially resolved:**

The finite-sample error bound via Matrix Bernstein does not fully answer Q1 as originally posed. Q1 asked whether minimising the empirical objective recovers functions close to the true singular functions. The bound provided controls the difference in loss values between $L_{\text{pop}}$ and $L_{\text{emp}}$, but closeness of loss values does not imply closeness of minimisers without additional conditions such as strong convexity or a spectral gap argument. This is neither clarified nor discussed.

**Unresolved:**

The worsening of FID for $K > 128$ was not addressed. Specifically, the mechanism by which increasing $K$ degrades sample quality remains unexplained.

**Conditions for acceptance:**

I ask that acceptance be conditioned on the following: full proofs being included in the final manuscript as promised; a clear explanation of the FID/K trade-off; a direct response to Q4 reconciling the conceptual incompatibility with inverse problems against the use of DPS as a baseline, acknowledged in the limitations; and a mathematically rigorous exposition of the finite-sample error in the variational objective of Theorem 4.2.

**Key Questions For Authors:**

1. Theorem 4.2 holds for when the exact expectation can be calculated, what happens when working with finite-sample versions? Can the error between the finite-sample variant and the true expectation be quantified?
2. Algorithm 1 applies a stop-gradient operation making the training procedure asymmetric and disconnecting it from the symmetric variational objective in Theorem 4.2. Does removing the stop-gradient cause training divergence or merely slower convergence?How does it affect the quality of the learnt singular functions? (Weakness 5)
3. How would the method fare for higher resolution images? Can this be applied in the latent space of a VQ-VAE or Stable Diffusion architecture?
4. Can this be extended to conditional sampling in the context of linear inverse problems in imaging?

**Limitations:**

The authors do not include a limitations section. The dependence on access to training data during deployment, and the gap between the theoretical objective and the stop-gradient training procedure are not acknowledged as limitations anywhere in the paper. These must be addressed explicitly.

The authors must rewrite the impact statement to accurately reflect geniune concerns that could arise from the misuse of generative modelling and the effect of biases of datasets.

**Strengths And Weaknesses:**

__Strengths:__
1. Leveraging a low-rank approximation of the conditional expectation with operator theory is novel and intriguing. The derivation showing the Rayleigh-Ritz characterization of the top-K eigenvalues of the covariance operator provides a neat, first-principles justification to use diffusion-based forward samples as positives.
2. The unification of label, CLIP and mask guidance under a single task-agnostic framework without retraining is elegant and practically useful.
3. The spectral phase transition analysis is an interesting contribution that allows for an insightful visualization of the window where guidance would serve useful.
4. The CIFAR-10 accuracy of 89.4% compared to the 52.0% for TFG is clearly a marked improvement due to the proposed method.

__Weaknesses:__
1. Compactness of the $T_{t}T^{*}_{t}$ is not obvious and must be proved.
2. The claim that "At high noise levels, most singular values of $T_t$ are small." must be supported with some evidence.
3. Section 5 is hard to read and can be articulated better.
4. Experiments are restricted to CIFAR-10 ($32\times 32$) and CelebA-HQ ($256\times 256$) images.
5. The stop-gradient trick is introduced as an empirical method to stabilize training and is not sufficiently motivated by any theory. The authors must provide an ablation by showing the effect of removing the stop-gradient.
6. The guidance formulae are truncated in practice and this is clearly an approximation of the infinite sum. This error must be theoretically characterized for a thorough understanding of the effect of the terms being neglected.
7. Fig. 5(b) shows that FID worsens for values of $K$ beyond $128$ even as accuracy continues to improve. This trade-off is not discussed anywhere in the paper and represents an important practical consideration for choosing $K$.

Minor Issues:
1. In line 64, "37 pp" is mentioned, this could be written as 37%.

---

> ### Author Rebuttal · Authors · 2026-03-31
>
> We thank the reviewer for the constructive feedback and theoretical questions.
>
> **Compactness (W1).** Compactness is currently addressed in Appendix A.2. We will explain this result more carefully. To prove $T_t$ is compact, we show that it is Hilbert-Schmidt i.e., $\\|T_t\\|_\mathrm{HS}^2 = \iint \frac{p(x_0, x_t)^2}{p_0(x_0) p_t(x_t)} dx_0 dx_t < \infty$. For DDPMs, $p(x_t \mid x_0)$ is Gaussian. Because images consist of bounded pixel values, $p_0(x_0)$ has compact support. Consequently, the marginal $p_t(x_t) = \int p(x_t \mid x_0)p_0(x_0)dx_0$ is a Gaussian mixture over a bounded domain. This guarantees that the tails of $p(x_t \mid x_0)$ and $p_t(x_t)$ decay at the same exponential rate. Because the denominator $p_t(x_t)$ does not decay faster than the numerator, the density ratio remains bounded, and the integral is finite for $t > 0$.
>
> **Vanishing Singular Values (W2).** We will update Appendix A.3 with a derivation of the following results (shown in Figs. 6(a) and 6(b)):
> - $T\_t$ is a contraction $\\|T\_t f\\|\_{L^2(\mu\_t)} \leq \\|f\\|\_{L^2(\mu\_0)}$. Thus, the singular values are always bounded above by 1.
> - $\\|T_t f\\|_{L^2(\\mu_t)} \to 0$, as $\\bar{\\alpha}_t \to 0$. Thus, singular values decay to zero.
>
> **Readability of Section 5 (W3).** We appreciate the feedback. We will reformulate Section 5 to improve clarity, by opening the section with a summary of the offline/online stages.
>
> **Limited Experiments (W4).** We have conducted an additional experiment, by training our $f_\phi$ network with $K=2000$ (ridge $\xi=10^{-3}$) on ImageNet, following Algorithm 1. We report the guidance results for the 4 labels considered in the TFG paper:
>
> **Table: ImageNet Guidance Results**
> |Method| DPS|LGD|FreeDoM|MPGD|UGD|TFG|Ours|
> |-|-|-|-|-|-|-|-|
> | Accuracy (%)|38.8|11.5| 19.7| 6.80 | 25.5 | 40.9 | **41.6**|
> | FID ↓ | 193| 210| 200|239| 205|176|**183**|
>
> Two points are worth noting. First, TFG is a framework that subsumes all others under a unified hyperparameter search. Matching TFG therefore means matching the best combination of all training-free methods. Second, the spectral transition on ImageNet (like that observed for CIFAR-10 and CelebA-HQ in Fig. 6) occurs in the final 15% of timesteps of the forward diffusion trajectory. Within this short window, matching the best combined baseline confirms that the representational quality of ​$f_\\phi$ scales to complex datasets.
>
> **Stop-Gradient Ablation (W5, Q2).** Stop-gradient on one view is often used in SSL (SimSiam, BYOL) to prevent collapse. In our setting, the stop-gradient ensures stability and reduced memory overhead: the whitening operation requires an SVD, which has $\mathcal{O}(K^3)$ complexity. Backpropagating through the SVD is both memory-intensive and ill-conditioned at large $K$. Computing the whitening matrix on the stopped view bypasses this gradient.
> We will add the requested ablation below to the paper:
> **Table: Stop-Gradient Ablation on CIFAR-10**
> | |Accuracy (%)|FID|
> |-|-|-|
> |+Stop Grad|89.4|70.7|
> |−Stop Grad|80.4|51.6|
>
> Removing the stop-grad reduces accuracy. The FID improvement does not reflect better generation quality, but rather, worse guidance, which keeps the distribution closer to the prior.
>
> **Error Bounds (W6, Q1).** We will update the paper with Appendix A.5 Error Bounds:
> - The SVD truncation error is bounded by the largest discarded singular value. For a clean-data signal $h(x_0)$, the error between the posterior of $h$ given $x\_t$, denoted $T\_t h$, and our rank-$K$ approximation is
> $\\|T\_t h - T\_{t,K} h\\|_{L^2(\mu_t)}^2 \leq \\|h\\|\_{L^2(\mu\_0)}^2 \sigma^2\_{t,K+1}$.
> - The cosine of the principal angles between the true and the Monte-Carlo estimated singular spaces is bounded via the Davis-Kahan $\\sin\\Theta$ Theorem. For a sample of size of $B$, a covariance spectral gap $\\gamma\_K=\\lambda\_{K}-\\lambda\_{K+1}$, and assuming $\\|f\_\\phi(X,t)\\| < M$, we have with probability $1-\\delta$:
> $\|\\sin\Theta(\mathcal{U}\_K,\tilde{\mathcal{U}\_K})\\| \\leq 2\\frac{\\|C_t-\\tilde{C}_t\\|}{\\gamma_K} = \\mathcal{O}\\left(\\frac{M^2}{\\gamma\_K}\sqrt{\\frac{2\\log(K/\\delta)}{B}}\\right) $,
> where $\\mathcal{U}\_K$ and $\\tilde{\\mathcal{U}}\_K$ are the true and the estimated $K$-leading eigenspaces.
>
> **FID worsening for increasing $K$ (W7).** We thank the reviewer for this remark, which we will expound in the revised paper. The leading singular functions capture smooth, robust features. The higher-order functions (large $K$) capture noise-sensitive details. If we use too large a $K$, the guidance relies on higher-frequency details, which forces the model to perfectly satisfy the condition (high accuracy) at the cost of pushing the sample off the image manifold (worse FID).
>
> **Latent Space Formulation (Q3).** The framework extends naturally to latent diffusion. $f_\phi$​ would operate on the $4\times 64\times 64$ latent space, which is smaller than CelebA-HQ pixels, making training comparably lightweight.

---

> > ### Author Rebuttal · Reviewer_Dri7 · 2026-04-03
> >
> > Thank you for your response. I think the manuscript can be improved by addressing the following issues more carefully and thoroughly:
> >
> > - $T_t T^{*}\_t$ being compact relies on showing that it is a Hilbert-Schmidt operator (Appendix A.2), but the authors do not formally prove that the symmetric kernel of the operator is Hilbert-Schmidt. A complete, rigorous proof in the manuscript is necessary to establish this claim.
> >
> > - For vanishing singular values, the rebuttal merely states that the operator is a contraction without a proof sketch. Additionally, the claim that the norm tends to $0$ as $\bar{\alpha}_t \to 0$ is not supported by a proof sketch and is not obvious. This requires stronger justification.
> >
> > - The SVD truncation bound is stated without a proof sketch. It is a direct consequence of the Eckart-Young theorem, which could have been mentioned explicitly. Additionally, the bound is only informative for small values of $\sigma_{t,K+1}$, and it is not obvious why this holds in general.
> >
> > - The Davis-Kahan bound is stated without a proof sketch or derivation. Furthermore, it does not answer the original question regarding the finite-sample error in the variational objective of Theorem 4.2. Please provide a complete response to this question.
> >
> > - Regarding the worsening of FID for $K > 128$, the explanation is imprecise and vague. It is not clear how or why samples get pushed off the image manifold specifically as a consequence of increasing $K$.
> >
> > - The question regarding the extension of this framework to conditional sampling in the context of linear inverse problems (Q4) has not been addressed in this rebuttal. The response provided to Reviewer ptT8 on this point arguing that the low-rank approximation is incompatible with inverse problems due to the high-rank nature of the required posterior is not clear. Please provide a direct, clear response to this question and specify what exactly breaks down in that setting. This clarification is necessary since DPS is being used as a baseline.

---

> > > ### Author Response · Authors · 2026-04-07
> > >
> > > We sincerely thank the reviewer. We'll update the paper with full proofs. Due to character limit we can only provide sketches below.
> > >
> > > **Compactness.** Assume $p_0$ has compact support $\mathcal{X}$ and $\\|x_0\\|\leq R$.  We show that
> > >
> > > $\\|T\_t\\|\_{HS}^2=\\int_\\mathcal{X}p_0(x\_0)\\left(\\int\_{\\mathbb{R}^D}\\frac{p\_t(x\_t\\mid x\_0)^2}{p\_t(x\_t)}dx\_t \right)dx\_0<\\infty$
> > >
> > > The numerator is
> > >
> > > $p\_t(x\_t\\mid x\_0)^2=\\frac{1}{(2\\pi(1-\bar{\alpha}_t))^D}\\exp\\left(-\\frac{2\\|x\_t-\\sqrt{\\bar{\\alpha}\_t}x\_0\\|^2}{2(1-\\bar{\\alpha}\_t)}\\right)$
> > >
> > > Write $p\_t(x\_t)=\\int\_\\mathcal{X}p\_t(x\_t\\mid y)p(y)dy$. Using $\\|x\_t-\\sqrt{\\bar{\\alpha}\_t} y\\|^2 \\le (\\|x_t\\|+\\sqrt{\\bar{\\alpha}\_t} R)^2$,
> > >
> > > $p\_t(x\_t)\\ge\\frac{1}{(2\\pi(1-\\bar{\\alpha}\_t))^{D/2}}\\exp\\left(-\\frac{\\|x\_t\\|^2+2\\sqrt{\\bar{\\alpha}\_t}R\\|x\_t\\|+\\bar{\\alpha}\_t R^2}{2(1-\\bar{\\alpha}\_t)}\\right)$
> > >
> > > Thus,
> > >
> > > $\\frac{p\_t(x\_t\\mid x\_0)^2}{p\_t(x\_t)}\\le\\frac{1}{(2\\pi(1-\\bar{\\alpha}\_t))^{D/2}}\\exp\\left(\\frac{-\\|x_t\\|^2+ 4\\sqrt{\\bar{\\alpha}\_t}x\_t^\\top x\_0+2\\sqrt{\\bar{\\alpha}\_t}R\\|x\_t\\|-2\\bar{\\alpha}\_t\\|x_0\\|^2+\\bar{\\alpha}\_t R^2}{2(1-\\bar{\\alpha}\_t)}\\right)$
> > >
> > > From Cauchy-Schwarz, $x\_t^\\top x_0\\le R\\|x\_t\\|$. Dropping $-2\\bar{\\alpha}\_t\\|x_0\\|^2\\le 0$
> > >
> > > $\text{Exponent}\\le\\frac{-\\|x\_t\\|^2+6\\sqrt{\\bar{\\alpha}\_t}R\\|x\_t\\|+\\bar{\\alpha}\_t R^2}{2(1-\\bar{\\alpha}\_t)}$
> > >
> > > We have $\\int\_{\\mathbb{R}^D}\\frac{p\_t(x\_t\mid x\_0)^2}{p\_t(x\_t)} dx\_t\\le  C(t,R)$. Since the leading term of the exponent is $-\\|x\_t\\|^2$, the integrand has Gaussian tails and $C(t,R)<\\infty$. Substituting, $\\|T\_t\\|_{HS}^2\\le C(t,R)\int\_\mathcal{X}p(x\_0)dx\_0=C(t,R)<\infty$. Thus $\\|T_t\\|\_{HS}^2<\infty$, hence compact. The adjoint $T_t^\ast$ and the composition $T\_tT\_t^\\ast$ are then compact too.
> > >
> > > **$T_t$ is a contraction.** Follows from Jensen's inequality.
> > >
> > > **Vanishing Singular Values.** For a zero-mean $f\\in L^2(p_0)$,
> > >
> > > $(T\_t f)(x_t)=\int f(x_0)p\_t(x\_0\mid x_t)dx_0=\int f(x_0)\left( \frac{p_t(x_t \mid x_0)}{p_t(x_t)}-1\right)p_0(x_0)dx_0$
> > >
> > > Thus,
> > >
> > > $\\|T_t f\\|\_{L^2(p_t)}^2 \\le \\|f\\|\_{L^2(p_0)}^2 \\iint \\frac{(p\_t(x\_t \\mid x\_0) - p\_t(x\_t))^2}{p\_t(x\_t)} p_0(x\_0) dx\_0 dx\_t$   (Cauchy-Schwarz)
> > >
> > > The double integral is the $\chi^2$-divergence between $p(x\_0,x\_t)$ and $p_0(x\_0)p\_t(x\_t)$. Assume that $p_0$ has compact support. As $\\bar{\\alpha}\_t\\to 0$, $p\_t(x\_t \\mid x\_0)$ and $p\_t(x\_t)$ converge uniformly to $\\mathcal{N}(0,I)$. Thus, the $\\chi^2$-divergence and $\\|T_t f\\|\_{L^2(p\_t)}^2$ vanish. Since $\\sigma\_{t,1}=\\sup\_{\\|f\\|=1} \\|T\_t f\\|\_{L^2(p\_t)}$, we have $\\sigma\_{t,k}\\to 0$.
> > >
> > > **SVD Truncation.** We’ll explicitly mention Eckart-Young. Indeed, the bound is informative for small $\\sigma\_{t,K+1}$. However, empirically (Fig 6) and from the result above, the singular values decay to 0. Thus, for large enough $t$ (guidance windows from Sec 6.5) the approximation is well suited.
> > >
> > > **Finite Sample Error.** Let $f\in\mathbb{R}^K$ be the network output and assume $\\|f(X\_t)\\|_2\\leq M$. Let the population covariance be $C=\\mathbb{E}[f(X\_t)f(X\_t)^\\top]$ and cross-covariance $\\tilde{C}=\\mathbb{E}[f(X\_t)f(\\tilde{X}\_t)^\\top]$. Let $\\hat{C}$ and $\\hat{\tilde{C}}$ be their empirical counterparts (batch size $B$). The objectives (ridge $\xi$) are
> > >
> > > $L\_{\\text{pop}}=\\mathrm{Tr}((C+\\xi I)^{-1/2} \tilde{C}(C+\\xi I)^{-1/2}), \\quad L\_{\\text{emp}} = \\mathrm{Tr}((\\hat{C}+\\xi I)^{-1/2} \\hat{\\tilde{C}} (\\hat{C}+\\xi I)^{-1/2})$
> > >
> > > Let $Z_k=f(X\_t^{(k)})f(X\_t^{(k)})^\\top$ and $S_k=\\frac{1}{B}(Z\_k-C)$.
> > >
> > > - The covariance error is $Z=\\sum_k S_k$, which we bound via 6.1.1 (Matrix Bernstein) from Tropp (2015), using
> > >
> > > $\\|S_k\\|_2\\leq\\frac{1}{B}(\\|Z_k\\|_2+\\|C\\|_2)\leq\frac{2M^2}{B}$
> > >
> > > and
> > >
> > > $v(Z)=\\left\\|\\sum\_{k=1}^B\\mathbb{E}[S_k^2] \\right\\|_2\\le\\left\\|\\sum\_{k=1}^B\\frac{M^2}{B^2}C\\right\\|\_2=\\frac{M^2}{B}\\|C\\|_2\\le\\frac{M^4}{B}$
> > >
> > > This yields $\\|C-\\hat{C}\\|_2\\le\\mathcal{O}\left(M^2\\sqrt{\\frac{\\log(K/\delta)}{B}}\\right)$. The same bound follows for $\\|\tilde{C}-\\hat{\tilde{C}}\\|_2$.
> > >
> > > - Using the Lipschitz continuity of the inverse square root
> > >
> > > $\\|(\\hat{C}+\\xi I)^{-1/2}-(C+\\xi I)^{-1/2}\\|_2\le\frac{1}{2\xi^{3/2}}\\|C-\hat{C}\\|\_2\\le\\mathcal{O}\\left(\frac{M^2}{2\\xi^{3/2}}\\sqrt{\\frac{\\log(K/\delta)}{B}}\\right)$
> > >
> > > - Using the triangle inequality and $|\\mathrm{Tr}(A)|\\le K\\|A\\|_2$,
> > >
> > > $|L_{\text{pop}}-L_{\text{emp}}| \le K(\\|(C+\xi I)^{-1/2}\\|_2^2\\|\tilde{C}-\hat{\tilde{C}}\\|_2+2\\|(C+\xi I)^{-1/2}\\|_2 \\|\hat{\tilde{C}}\\|_2\\|(C+\xi I)^{-1/2}-(\hat{C}+\\xi I)^{-1/2}\\|_2)$
> > >
> > > Substituting, we get
> > >
> > > $|L_{\text{pop}}-L_{\text{emp}}|\le\mathcal{O}\left( K M^2 \left( \frac{1}{\xi}+\frac{M^2}{\xi^{2}}\right)\sqrt{\frac{\log(K/\delta)}{B}}\right)$
> > >
> > > with probability at least $1-\\delta$.
> > >
> > > **Inverse Problems.** We refer to the reply provided to Reviewer ptT8 due to lack of space.

---

### Official Review · Reviewer_9yED · 2026-03-15

**Soundness:** 4
**Presentation:** 3
**Significance:** 3
**Originality:** 4
**Overall Recommendation:** 5
**Confidence:** 4

**Summary:**

The paper applies the spectral decomposition over the posterior mean operator with any guidance function. The decomposition generates numbers of singular functions, which served as feature coordinates for image manifold. Furthermore, the paper proposed a self-supervised learning algorithm to learn these singluar functions under different guidance functions. Experiment results demonstrate the effectivenss and efficiency of the proposed methods compared with many previous works over different datasets and different guidance task.

**Compliance With Llm Reviewing Policy:**

Affirmed.

**Final Justification:**

The rebuttal has addressed my main concern regarding scalability. Therefore, I believe this is a solid paper and am inclined to accept it.

**Key Questions For Authors:**

I have no more questions for the paper, however I have some suggestions to the writting of the paper. I spend a lot of time undersanding the section 4 of this paper. The spectral decomposition over a operator (equation 9) is not intuitive for me to understand. It is better to add some background or provide some intuitions for better understanding this part.

**Limitations:**

Yes

**Strengths And Weaknesses:**

# Strength

1. The proposed method is novel and clearly motivated. Through spectral decomposition, at each timestep, we could obtain the feature coordinate for the image manifold. And through optimization problem defined in Theorem 4.2, we could use self-supervised learning method to learn these coorindate. And with those coorindates, we could achieve guided generation.
2. The experiments are solid and convincing. The paper uses different datasets, guidance task, compared with different baseline methods for support their methods. There are also ablation stuidies for key factors in their methods.


# Weakness

1. Limited scale. While the proposed method has good performance on unconditional CIFAR-10, CelebA diffusion models, it is unknown whether the method will work for large scale text-to-image diffusion model. For example, will the method work for mask guidance on Stable Diffusion?
2. Lack some related works. There are other works exploring using self-supervised learning for conditional generation [1] and applies spectral decomposition over posterior mean to obtain feature coordinate and achieve conditional generation [2, 3].

[1] Dalva, Yusuf, and Pinar Yanardag. "Noiseclr: A contrastive learning approach for unsupervised discovery of interpretable directions in diffusion models." In Proceedings of the IEEE/CVF conference on computer vision and pattern recognition, pp. 24209-24218. 2024.

[2] Park, Yong-Hyun, Mingi Kwon, Jaewoong Choi, Junghyo Jo, and Youngjung Uh. "Understanding the latent space of diffusion models through the lens of riemannian geometry." Advances in Neural Information Processing Systems 36 (2023): 24129-24142.

[3] Chen, Siyi, Huijie Zhang, Minzhe Guo, Yifu Lu, Peng Wang, and Qing Qu. "Exploring low-dimensional subspace in diffusion models for controllable image editing." Advances in neural information processing systems 37 (2024): 27340-27371.

---

> ### Author Rebuttal · Authors · 2026-03-31
>
> We thank the reviewer for the careful reading and positive assessment. We address the two weaknesses and the writing suggestion below.
>
> **Limited Scale (W1).** To strengthen the empirical evaluation of our framework, we have conducted additional experiments on ImageNet, which contains a much greater semantic diversity. We followed the experimental setup used in CIFAR-10 and CelebA-HQ. We trained our $f_\\phi$ network with $K=2000$, following Algorithm 1 (ridge $\\xi=10^{-3}$). We report the guidance benchmark results for the 4 ImageNet labels considered in the TFG paper:
>
> **Table: ImageNet Guidance Results**
> | Method           | DPS  | LGD  | FreeDoM | MPGD | UGD  | TFG  | Ours |
> | ---------------- | ---- | ---- | ------- | ---- | ---- | ---- | ---- |
> | **Accuracy (%)** | 38.8 | 11.5 | 19.7    | 6.80 | 25.5 | 40.9 | **41.6**    |
>  | **FID ↓**             | 193  | 210  | 200     | 239  | 205  | 176  | **183**   |
>
> Two points are worth noting. First, TFG is a framework that subsumes all others under a unified hyperparameter search. Matching TFG therefore means matching the best achievable combination of all training-free methods. Second, the spectral transition on ImageNet (like that observed for CIFAR-10 and CelebA-HQ in Fig. 6) occurs in the final 15% timesteps of the forward diffusion trajectory. Within this short window, matching the best combined baseline confirms that the representational quality of $f_\phi$​ scales to complex datasets.
>
> Also note that the framework is directly applicable to latent diffusion models: $f_\\phi$​ would operate on the latent space rather than pixel space, and the diffusion pairs $x_t$, $\\tilde{x}\_t$ would be drawn from the latent forward process. SD's latent space is $4 \\times 64 \\times 64$, which is actually smaller than CelebA-HQ pixels, so training $f_\\phi$​ would be comparably lightweight. We leave a full SD evaluation to future work but note that the theoretical framework requires no modification.
>
> **Additional Works (W2).** We are thankful for the provided references, which we will include in the related work and situate compared to our approach:
>
> NoiseCLR [1] discovers interpretable directions in diffusion noise space via contrastive learning. While it shares the use of SSL-style objectives, it is designed for latent space editing rather than guidance, and it does not characterize which directions are information-preserving across noise levels, which the spectral decomposition of $T_t​ T_t^\\ast$​ provides.
> Park et al. [2] and Chen et al. [3] are more closely related. Both apply spectral decomposition to statistics derived from the posterior mean $\\mathbb{E}[X_0 \\mid x_t]$, treating it as a post-hoc tool for understanding latent geometry and finding editing directions. Our work differs in three fundamental ways. First, we decompose the conditional expectation operator​ directly, whose SVD characterizes the structure of the diffusion process, rather than a statistic derived from denoiser outputs. Second, Theorem 4.2 provides a variational characterization of the leading singular functions that connects directly to SSL training, enabling us to learn these functions rather than computing them post-hoc. Third, our framework yields a complete guidance algorithm with theoretical guarantees, whereas [2] and [3] are primarily analytical tools without a guidance mechanism.
>
> **Writing / Background on Spectral Decompositions (Q1).** We appreciate the feedback and will revise this section to provide further intuition. If the data space were finite and discrete, $T_t$ would simply be a transition probability matrix mapping clean states to noisy states. Equation 9 would then be the standard Singular Value Decomposition (SVD) of that matrix. The singular values quantify how much information about the clean state survives the noise corruption with large singular values corresponding to recoverable structure and small ones to details erased by the forward process. The continuous formulation simply generalizes this to $L^2$ function spaces. We will revise Section 4 to include the finite/discrete intuition as a motivating example before introducing the operator formalism.

---

> > ### Author Rebuttal · Reviewer_9yED · 2026-04-04
> >
> > Thank you for your responses. I have increased my score to 5.

---

> > > ### Author Response · Authors · 2026-04-07
> > >
> > > We sincerely thank the reviewer for the thoughtful feedback and for taking the time to engage with our paper. We are glad that the additional experiments and clarifications addressed your concerns.
> > >
> > > We will incorporate the discrete/finite intuition for the operator in Equation 9, along with additional explanation to make the spectral decomposition more accessible. We will also include the suggested related works and clearly position our contributions with respect to them.

---

### Decision · Program_Chairs · 2026-04-30

**Decision:**

Accept (regular)

**Comment:**

This paper introduces Spectral Guidance, a framework that decomposes the conditional expectation operator of the diffusion process via its singular functions, learns them through a self-supervised objective, and projects arbitrary guidance signals onto the sampling trajectory. Scores are 5/4/4/4, with all reviewers positive.

The theoretical contribution is novel and well-grounded: the variational characterization (Theorem 4.2) provides a principled SSL training objective, and the phase transition analysis identifies optimal guidance windows. The method is versatile: a single task-agnostic network supports label, CLIP, and mask guidance without retraining or auxiliary models, with ~4x faster sampling than TFG. CIFAR-10 accuracy improves by 37pp over the strongest training-free baseline.

During rebuttal, Reviewer 9yED's scalability concern was addressed with ImageNet experiments (fully resolved, raised to 5). Reviewer Dri7 requested formal proofs; sketches were provided with full versions promised for camera-ready. Reviewer ptT8 asked about inverse problems; the authors gave a clear explanation of why the low-rank projection targets semantic rather than pixel-level guidance. Reviewer QQaf noted marginal ImageNet gains over TFG, but the authors achieved these with zero dataset-specific tuning against TFG's optimized hyperparameter search.

The authors should incorporate all promised proofs, ImageNet experiments, Gaussian validation, and an explicit limitations section into the camera-ready.